# Learning to Draw: Emergent Communication through Sketching

**Daniela Mihai**
Electronics and Computer Science
The University of Southampton
Southampton, UK
adm1g15@soton.ac.uk

**Jonathon Hare**
Electronics and Computer Science
The University of Southampton
Southampton, UK
jsh2@soton.ac.uk

## Abstract

Evidence that visual communication preceded written language and provided a basis for it goes back to prehistory, in forms such as cave and rock paintings depicting traces of our distant ancestors. Emergent communication research has sought to explore how agents can learn to communicate in order to collaboratively solve tasks. Existing research has focused on language, with a learned communication channel transmitting sequences of discrete tokens between the agents. In this work, we explore a visual communication channel between agents that are allowed to draw with simple strokes. Our agents are parameterised by deep neural networks, and the drawing procedure is differentiable, allowing for end-to-end training. In the framework of a referential communication game, we demonstrate that agents can not only successfully learn to communicate by drawing, but with appropriate inductive biases, can do so in a fashion that humans can interpret. We hope to encourage future research to consider visual communication as a more flexible and directly interpretable alternative of training collaborative agents.

## 1 Introduction

Imagine you and a friend are playing a game where you have to get your friend to guess an object in the room by you sketching the object. No other communication is allowed beyond the sketched image. This is an example of a *referential communication game*. To play this game you need to have learned how to draw in a way that your friend can understand. This paper explores how artificial agents parameterised by neural networks can learn to play similar drawing games. More specifically, we reformulate the traditional referential game such that one agent draws a sketch of a given photo and the second agent guesses, based on the drawing, the corresponding photo from a set of images.

Spurred by innovations in artificial neural networks, deep and reinforcement learning techniques, recent work in multi-agent emergent communication [4, 17, 19, 29, 37] pursues interactions in the form of gameplay between agents to induce human-like communication. Artificial communicating agents can collaborate to solve various tasks: image referential games with realistic visual input [19, 28, 29], negotiation [2], navigation of virtual environments [7, 22], reconstruction of missing input [4, 17] and, more recently, drawing games [11]. The key to achieving the shared goal in many of these games is collaboration, and implicitly, communication. To date, studies on communication emergence in multi-agent games have focused on exploring a language-based communication channel, with messages represented by discrete tokens or token sequences [6, 17, 19, 25, 28, 29, 35]. However, these communication protocols can be difficult for a human to interpret [3, 26, 33]. In this work we propose a direct and potentially self-explainable means of transmitting knowledge: *sketching*.

Evidence suggests pre- and early-humans were able to communicate by drawing long before developing the various stages of written language [20, 38]. Drawings such as petrograms and petroglyphs

35th Conference on Neural Information Processing Systems (NeurIPS 2021).

exist from the oldest palaeolithic times and may have been used to record past experiences, events, beliefs or simply the relation with other beings [13, 21]. These pictorial characters which are merely impressions of real objects or beings stand at the basis of all writing [16]. This leads us to question if drawing is a more natural way of *starting* to study emergent communication and if it could lead to better written communication later on.

This idea has recently gained interest in several domains. In the cognitive science literature, neural models of sketching have been developed to study the factors which enable contextual flexibility in visual communication [8]. Likewise, works such as Fernando et al. [11] attempt to automate the artistic process of drawing by training agents in a reinforcement learning framework, to play a variety of drawing games. However, the focus of this paper is to open the doorway to exploring different types of communication between artificial agents and humans. The novelty of our work is also evident in the model framework which can be easily extended well beyond aspects previously studied.

Concretely, we propose a visual communication channel in the context of image-based referential games. We leverage recent advances in differentiable sketching that enables us to construct an agent that *can learn to communicate intent through drawing*. Through a range of experiments we show that:

- Agents can successfully communicate about real-world images through a sketching game. However, training with a loss that tries to maximise gameplay alone does not lead to human decipherable sketches, irrespective of any visual system preconditioning;
- Introducing a perceptual loss improves human interpretability of the communication protocol, at little to no cost in the gameplay success;
- Changes to the game objective, such as playing an object-oriented game, can steer the emergent communication protocol towards a more pictographic or symbolic form of expression;
- Inducing a shape-bias into the agents' visual system leads to more explainable drawings;
- A drawing agent trained with a perceptual loss can successfully communicate and play the game with a human.

## 2 Communication between agents

Communication emerges when two or more participants are involved, share a goal, task or incentive which can be achieved only by transfer of information and so is beneficial for all parties involved. Studies on language origins [36, 41] consider cooperation to be a key prerequisite to language evolution as it implies multiple agents having to self-organise and adapt to the same convention. Studies on the emergence of communication in cooperative multi-agent environments from recent years have focused on (natural) language learning [28, 29] and its inherent properties such as compositionality and expressivity [17, 18, 37].

A number of works specifically relate to the overarching ideas of gameplay and learning in this paper. For example, Foerster et al. [12] proposed a framework for differentiable training of communicating agents which was later used by Jorge et al. [23] to solve image search tasks with two interacting agents communicating with atomic symbols. Lazaridou et al. [28] proposed an image-based referential game in which the agents again communicated using atomic symbols, and were trained using policy gradients. Havrylov and Titov [19] and Mordatch and Abbeel [35] both demonstrated that it was possible to use differentiable relaxations to train agents that communicated with sequences of symbols. In the former case, the agents played the referential game that we adopt for our experiments.

One of the long-term goals of this research in language emergence is to develop interactive machines that can productively communicate with humans. As such we should ensure that whatever language artificial agents develop, it is one that human agents can understand. In our work, we take inspiration from the process and evolution of writing. Written language has undergone many transitions from early times to reach the forms we now know: from pictures and drawings to word-syllabic, syllabic and, finally, alphabetic systems. In the beginning, our early ancestors did not know how to communicate in writing. Instead, they began drawing and painting pictures of their life, representing people and things they knew about [16]. Studies on the communication systems developed in primitive societies compare ancient drawings to the very early sketches drawn by children and talk about their tendency of concretely identifying certain things or events in their surrounding world [16, 24]. Psychological and behavioural studies have shown that children try to communicate to the world through the images they create even when they cannot associate them with words [9].

# 3 A model for learning to communicate by drawing

We present a model consisting of two agents, the sender and the receiver in which the sender learns to draw by playing a game with the receiver. The overall architecture of the agents in the context of the game they are learning to play is shown in Figure 1. Full code for the model and all experiments can be found at `https://github.com/Ddaniela13/LearningToDraw`.

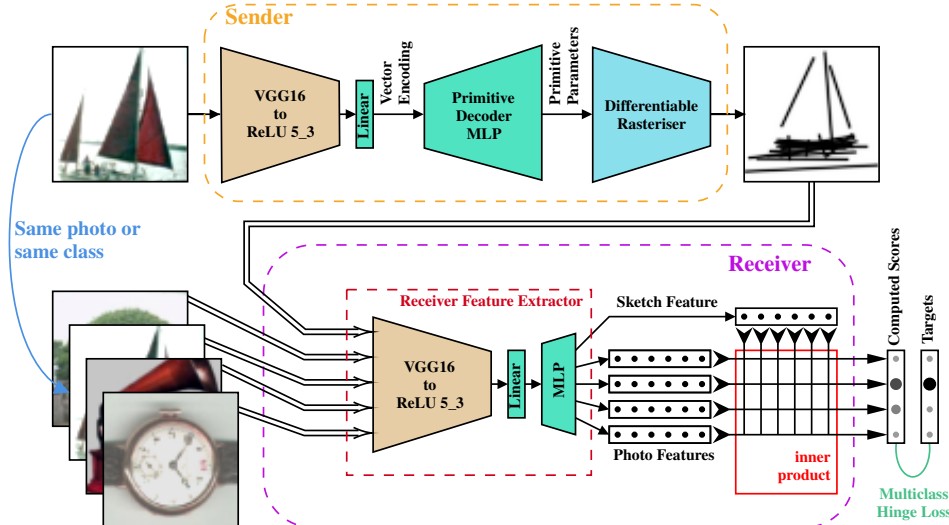

Figure 1: **Overview of the agent architecture and game setup.** The 'sender' agent is presented with an image and sketches its content through a learnable drawing procedure. The 'receiver' agent is presented with the sketch and a collection of photographs, and has to learn to correctly associate the sketch with the corresponding photograph by predicting scores which are compared to a one-hot target. Both agents are parameterised by neural networks trained end-to-end using gradient methods.

## 3.1 The Game Environment

Our experimental setup builds upon the image referential game previously explored in studies of emergent communication [19, 28, 29] that derives from Lewis's signalling game [30]. We implemented several variants of Havrylov and Titov [19]'s image guessing game. The overall setting of these games is formulated as follows:

1. Two target photographs, $\mathbf{P}_s$ and $\mathbf{P}_r$, and set of $K$ distractor photographs, $\{\mathbf{P}_d^{(k)}\}_{k=1}^K$, are selected.
2. There are two agents: a sender and a receiver.
3. After being presented the $\mathbf{P}_s$ target image, the sender has to formulate a message conveying information about that image.
4. Given the message and the set of photographs, $\{\mathbf{P}_d^{(k)}\}_{k=1}^K \cup \{\mathbf{P}_r\}$, consisting of all the distractors and the target $\mathbf{P}_r$, the receiver has to identify the target correctly.

The specifics of how the photographs are selected (step 1 above) depend on the game variant as described below. Success in these games is measured by the binary ability of the receiver to correctly guess the correct image or not; as such, the measure of *communication rate* is used to assess averaged performance over many games using independent images to those used during training. Unlike Havrylov and Titov [19]'s game in which the sender helps the receiver identify the correct image by sending a message constructed as a sequence of tokens drawn from a predefined vocabulary, we propose using a directly interpretable means of communication: *sketching the target photograph*.

**Original game variant.** In Havrylov and Titov [19]'s variant of the game there is a pool of photos from which the distractors and target $\mathbf{P}_s$ are drawn randomly without replacement. The target $\mathbf{P}_r$ is set to be equal to $\mathbf{P}_s$. In our *original* variant experiments the number of distractors, $K$, is set to 99.

**Object-oriented game variants.** In addition to the original setup, we explored two slightly different and potentially harder game configurations which were intended to induce the agents to draw sketches that would be more representative to the object class they belong to rather than to the specific instance of the class. These setups use labelled datasets where each image belongs to a class based on its contents. In the first of these variants (we refer to this as *OO-game same*), the target $\mathbf{P}_r$ is set to be equal to $\mathbf{P}_s$, and the distractors and target are sampled such that their class labels are disjoint (that is every photo provided to the receiver has a different class). The second setup (*OO-game different*) is similar to the first, but the target $\mathbf{P}_r$ is chosen to be a different photograph with the same class label as target $\mathbf{P}_s$. The intention behind these games is to explore a universally interpretable depiction of the different object classes, which does not focus on individual details but rather conveys the concept. To some extent, this task is an example of multiple instance classification within a weakly supervised setting [1], which has been previously explored in the emergent communication literature [28].

## 3.2 Agents' Architectures

Both agents act on visual inputs. The agents are parameterised by deep neural networks and are trained using standard gradient techniques (Section 3.3).

**The agent's early visual system.** We choose to model the early visual systems of both agents with the head part of the VGG16 CNN architecture [40] through to the $\mathrm{ReLU}$ activation at the end of the last convolutional layer (commonly referred to as the `ReLU5_3` layer) before the final max-pooling and fully connected layers. In all experiments, we utilise pretrained weights and freeze this part of the model during training. We justify this choice on the basis that it provides the agents with an initial grounding in understanding the statistics of the visual world, and ensures that the visual system cannot collapse and remains universal. The weights are the standard `torchvision` ImageNet weights, except in the cases where we explore the effect of shape bias (see Section 4.5). As these pretrained weights were learned with images that were normalised according to the ImageNet statistics, all inputs to the VGG16 backbone (including sketches) are normalised accordingly. The output feature maps of this convolutional backbone are flattened and are linearly projected to a fixed dimensional vector encoding (64-dimensions unless otherwise specified). Because the datasets used in gameplay have different resolutions, the number of weights in the learned projection varies.

**Sender Agent.** The goal of the sender is to produce a sketch from the input photograph. For experiments in Section 4, we restrict the production of sketches to be a drawing composed of 20 black, constant width, straight lines on a white canvas of the same size as the input images. Experiments with fewer lines can be found in Appendix A. It is of course possible to have a much more flexible definition of a sketch and incorporate many different modelling assumptions. We choose to leave such exploration for future work and focus on the key question of whether we can actually achieve successful (and potentially interpretable) communication with our simplified but not unrealistic setup.

Given an input image, the agent's early visual system produces a vector encoding which is then processed by a three-layer multilayer perceptron (MLP) that learns to decode the primitive parameters used to draw the sketch. This MLP has $\mathrm{ReLU}$ activations on the first two layers and $\tanh$ activation on the final layer. Unless otherwise specified, the first two layers have 64 and 256 neurons respectively. The output layer produces four values for each line that will be drawn; the values are the start and end coordinates of each line stroke in an image canvas with the origin at the centre and edges at $\pm 1$.

To produce a sketch image from the line parameters output by the MLP, we utilise the differentiable rasterisation approach introduced by Mihai and Hare [34]. At a high level, this approach works by computing the distance transform on a pixel grid for each primitive being rendered. A relaxed approximation of a rasterisation function is applied to the distance transform to compute a raster image of the specific primitive. Finally, a differentiable composition function is applied to compose the individual rasters into a single image. More specifically, the squared Euclidean Distance Transform is computed, $\mathbf{D}_{\mathrm{seg}}^2(\boldsymbol{s}, \boldsymbol{e})$ over all pixels in the image, for each line segment starting at coordinate $\boldsymbol{s}$ and ending at $\boldsymbol{e}$. These squared distance transforms are simply images in which the value of each pixel is replaced with the closest squared distance to the line (computed when the pixels are mapped to the same coordinate system as the line — so the top left of the image is $(-1, -1)$ and bottom-right is

$(1, 1)$). Using the subscript $i$ to refer to the $i$-th line in the sketch, each $\mathbf{D}^2_{\text{seg}}(\boldsymbol{s_i}, \boldsymbol{e_i})$ is rasterised as

$$\mathbf{R}_i = \exp\left(-\frac{\mathbf{D}^2_{\text{seg}}(\boldsymbol{s_i}, \boldsymbol{e_i}))}{\sigma^2}\right) , \tag{1}$$

where $\sigma^2$ is a hyperparameter that controls how far gradients flow in the image, as well as the visible thickness of the line ($\sigma^2 = 5 \times 10^{-4}$ for all experiments in this paper). We adopt the soft-or composition function [34] to compose the individual line rasters into a single image, but incorporate an inversion so that a sketch image, $\mathbf{S}$, with a white canvas and black lines is produced,

$$\mathbf{S} = \prod_{i=1}^{n}(\mathbf{1} - \mathbf{R}_i) , \tag{2}$$

where $n$ is the number of lines. Finally, because the backbone CNNs work with three-band colour images, we replicate the greyscale sketch image three times across the channel dimension.

**Receiver Agent.** The receiver agent is given a set of photographs and a sketch image, and is responsible for predicting which photograph matches the sketch under the rules of the specific game being played. The receiver's visual system is coupled with a two-layer MLP with a ReLU nonlinearity on the first layer (the latter layer has no activation function). Unless otherwise specified, all experiments use 64 neurons in the first layer and 64 in the final layer. The sketch image and each photograph are passed through the visual system and MLP independently to produce a feature vector representation of the respective input. A score vector $\boldsymbol{x}$ is produced for the photographs by computing the scalar product of the sketch feature with the feature of each respective photograph. This score vector is un-normalised but could be viewed as a probability distribution by passing it through a softmax. The photograph with the highest score is the one predicted.

### 3.3 Training details

By incorporating a loss between the predicted scores of the receiver agent and the known correct target photograph, it is possible to propagate gradients back through both the receiver and sender agents. As such, we can train the agents to play the different game settings. For the loss function, we follow Havrylov and Titov [19] and choose to use Weston and Watkins [43]'s multi-class generalisation of hinge loss (aka multi margin loss),

$$\text{l}_{\text{game}}(\boldsymbol{x}, y) = \sum_{j \neq y} \max(0, 1 - \boldsymbol{x}_y + \boldsymbol{x}_j) , \tag{3}$$

where $\boldsymbol{x}$ is the score vector produced by the receiver, and $y$ is the true index of the target, and the subscripts indicate indexing into the vector. The rationale for this choice is that the (soft) margin constraint should help force the distractor photographs' features to be more dissimilar to the sketch feature. Tests using cross-entropy also indicated that it could work well as an alternative, however.

Optimisation of the parameters of both agents is performed using the Adam optimiser with an initial learning rate of $1 \times 10^{-4}$ for all experiments. For efficiency, we train the model with batches of games where the sender is given multiple images which are converted to sketches and passed to the receiver which reuses the same set of photographs for each sketch in the batch (with each sketch targeting a different receiver photograph). The order of the targets with respect to the input image's sketches is shuffled every batch. Batch size is $K + 1$, where $K$ is the number of distractors, for all experiments. Unless otherwise stated, training was performed for 250 epochs. A mixture of Nvidia GTX1080s, RTX2080s, Quadro RTX8000s, and an RTX-Titan was used for training the models. Higher resolution images required more memory. Training time varied from around 488 games/second (10 secs/epoch) for games using STL10 to around 175 games/second (around 5 mins/epoch) for Caltech-101 experiments with 128px images.

### 3.4 Making the sender agent's sketches more perceptually relevant

Perception of drawings has a long history of study in neuroscience [see e.g. 39, for an overview]. In order to induce the sender to produce sketches that are more interpretable, we explore the idea of using an additional loss function between the differences in feature maps of the backbone CNN

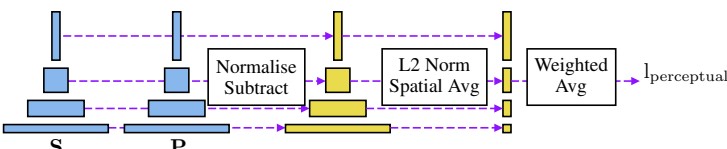

Figure 2: **Computing a 'perceptual' loss with the early visual system.** Features are extracted from the sketch $\mathbf{S}$ and corresponding photograph $\mathbf{P}$ from different layers of the backbone. The features are normalised over channels and subtracted. We take the sum of the squared differences over channels and average spatially. Finally, we compute a weighted average across layers.

from the produced sketch and the input image. Such a loss has a direct grounding in biology, where it has been observed through human brain imaging studies that sketches and photographs of the same scene result in similar activations of neuron populations in area V4 of the visual cortex, as well as other areas related to higher-order visual cognition [42]. At the same time, it has also been demonstrated that differences in feature maps from pre-trained CNN architectures can be good proxies for approximating human notions of perceptual similarity between pairs of images [45].

Inspired by Zhang et al. [45] we formulate a loss based on the normalised differences between feature maps of the backbone network from the application of the network to both the input photograph and the corresponding sketch. Unlike Zhang et al. we choose not to learn weightings for each feature map channel individually, but rather we consider all feature maps produced by a layer of the backbone to be weighted equally. Learning individual channel weighting would be an interesting direction for future research, but is challenging because we would want to avoid the network learning zero weights for each channel, where the perceptual loss is basically ignored.

Figure 2 illustrates our perceptual loss formulation; note also that unlike Zhang et al. [45] the final averaging operation does incorporate a (per-layer) weighting, $\boldsymbol{w}_l$, which we explore the effect of in Section 4.2. More formally, denoting the sketch as $\mathbf{S}$ and corresponding photo as $\mathbf{P}$, we extract $L = 5$ feature maps, $\hat{\mathbf{S}}^{(l)}, \hat{\mathbf{P}}^{(l)} \in \mathbb{R}^{H_l \times W_l \times C_l}$, for the $l$-th layer from the backbone VGG16 network and unit normalise each across the channel dimension. The loss is thus defined as,

$$\mathrm{l}_{\mathrm{perceptual}}(\mathbf{S}, \mathbf{P}, \boldsymbol{w}) = \sum_l \frac{\boldsymbol{w}_l}{H_l W_l} \sum_{h,w} \left\| \hat{\mathbf{S}}_{hw}^{(l)} - \hat{\mathbf{P}}_{hw}^{(l)} \right\|_2^2 . \tag{4}$$

To extract the feature maps we choose to use the outputs of the VGG16 layers immediately before the max-pooling layers (`relu1_2`, `relu2_2`, `relu3_3`, `relu4_3` and `relu5_3`). During training, this perceptual loss is added to the game loss ($\mathrm{l}_{\mathrm{game}}$). We note that the perceptual loss formulation is basically equivalent to the *content loss* in neural style transfer [14]. Neural style transfer combines this content loss with a *style loss* which encourages the texture statistics of a generated raster image to match a target image (which could be a sketch). Our model is different because instead of a loss *encouraging* a sketch-like style we directly *impose* production of sketches by drawing strokes.

## 4 Experiments

We next present a series of experiments where we explore if it is possible for the two agents to learn to successfully communicate, and what factors affect human interpretation of the drawings. We report numerical results averaged across 10 seeds for models evaluated on test sets isolated from training. Sample sketches from one seed are shown, but an overlay of 10 seeds can be found in Appendix J.

### 4.1 Can agents communicate by learning to draw?

We explore the game setups described in Section 3.1 and train our agents to play the games using $96 \times 96$ photographs from the STL-10 dataset [5]. For the *original* game we use 99 distractors. For the object-oriented games, due to the dataset only having 10 classes, we are limited to 9 distractors.

In Table 1, we show quantitative and qualitative results of the visual communication game played under the three different configurations. The results demonstrate that it is possible for agents to successfully play this type of image referential game by learning to draw. One can observe that although agents achieve a high communication success rate, using only the $\mathrm{l}_{\mathrm{game}}$ loss leads to the

Table 1: **Communication success rate and example sketches produced by the agents in order to achieve the game objective in various setups and with different losses.** Sample input images seen by the sender (the left column) are described as the sketches in the second and third column. Although successful communication seems to be achieved in all setups, the addition of the perceptual loss significantly improves human interpretability of the drawings. Examples are from STL-10.

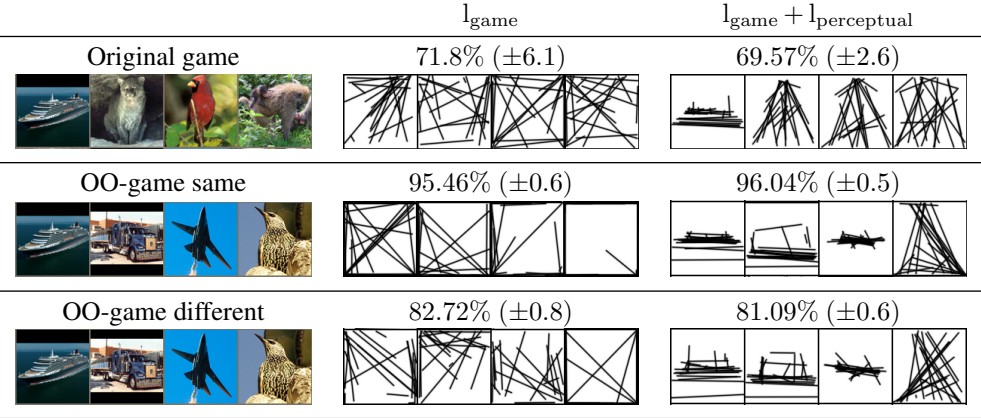

| | $l_{game}$ | $l_{game} + l_{perceptual}$ |
|---|---|---|
| Original game | 71.8% ($\pm$6.1) | 69.57% ($\pm$2.6) |
| OO-game same | 95.46% ($\pm$0.6) | 96.04% ($\pm$0.5) |
| OO-game different | 82.72% ($\pm$0.8) | 81.09% ($\pm$0.6) |

emergence of a communication protocol that is indecipherable to a human. However, the addition of the perceptual loss, motivated in Section 3.4, significantly improves the interpretability of the communication channel at almost no cost in the actual communication success rate.

One interesting observation is that although the sketches for some of the classes have greatly improved when incorporating the perceptual loss, for photographs of animals or birds, the sketches are not particularly representative of the class instance or distinguishable for the human eye. In the following sections we explore the model to try to better understand what factors affect drawing production.

## 4.2   What effect does weighting the perceptual loss have on the sketches?

Next, we explore the effect of manually weighting the perceptual loss. More precisely, we look at what happens when the perceptual loss is applied to the features maps from just one layer of the backbone network. As previously mentioned in Section 3.2, the feature maps are extracted using a VGG16 CNN up to `ReLU5_3` layer. For example, we can discard all feature maps except those from the first layer by weighting the perceptual loss by $[1, 0, 0, 0, 0]$. The effect of the different weights, which allow only one block of feature maps to be used for drawing the sketch, is illustrated in Table 2. We apply these constraints in two setups, the *original* and the *OO-game different*. In both cases, the drawings are unrecognisable if the perceptual loss takes into account only the first or the second block of feature maps. Blocks 3 through 5 seem to provide increasing structure under both game setups. It is worth noticing that, similar to the results shown in Section 4.1, the communication success rate in the original setup is always lower than that from the *OO-game different* setup. Overall, the information provided by individual layers in the visual extractor network is enough for the agents to develop a visual communication strategy that can be used to play the game. For humans, however, the later layers contribute the most to the emergence of a communication protocol that we can understand.

## 4.3   Does the *OO-game* influence the sketches to be more recognisable as the type of object?

Comparing the qualitative results of different game formats from Table 1, we notice that agents develop distinct strategies for representing the target photograph under different conditions. If there is more variability in the sketches that correspond to photographs from the same class in the original game setup, and a bit less in the *OO-game same*, the sketches become more like symbols representing all the photographs from one class when playing *OO-game different*. In other words, the object-oriented games influence the sketches to be more recognisable as the type of object, than the specific instance of the class. Further examples are shown in Appendix F.

Finally, it is worth noting how our results connect to how humans communicate through sketching when constrained under similar settings. The far/close contexts used in [8] are somewhat equivalent

Table 2: **The effect of weighting the perceptual loss such that only the feature maps from one backbone layer are used.** The features extracted in the last three layers of the visual system seem to capture information that leads to sketches which resemble to an extent the corresponding photograph.

| Loss weights | $[1, 0, 0, 0, 0]$ | $[0, 1, 0, 0, 0]$ | $[0, 0, 1, 0, 0]$ | $[0, 0, 0, 1, 0]$ | $[0, 0, 0, 0, 1]$ |
|---|---|---|---|---|---|
| Orig. game | 68.4% ($\pm3.6$) | 69.6% ($\pm2.2$) | 71.1% ($\pm2.4$) | 76.4% ($\pm2.1$) | 60.5% ($\pm4.8$) |
| |  |  |  |  |  |
| OO-game diff | 81.9% ($\pm1.2$) | 81.5% ($\pm0.9$) | 82.3% ($\pm0.9$) | 82.5% ($\pm0.5$) | 81.4% ($\pm0.8$) |
| |  |  |  |  |  |

Table 3: **The effect of the model's capacity on its sketches.** The wide model's sender encodes the photo into a 1024-dimensional vector (baseline 64), and the receiver's MLP linear layers have 1024 neurons each versus 64. Examples from training on Caltech-101 in the *OO-game different* setting.

| | Baseline | Wide |
|---|---|---|
| | 50.46% ($\pm1.5$) | 64.99% ($\pm1.5$) |
|  |  |  |

to our original/object-oriented settings. As Fan et al. [8] observe when humans play a similar drawing game, our agents achieve a higher recognition accuracy in settings that involve targets from different classes and develop different communication behaviours based on the context of the receiver.

### 4.4 How does the model's capacity influence the visual communication channel?

Regarding the model's architecture, we look into how drawings are influenced by the width of the model. In this experiment (results shown in Table 3), we compare the baseline model architecture detailed in Section 3.2 with a wider variant that has the following changes: the sender encodes the target photograph to a 1024-dimensional vector (baseline model encodes to 64-dimensional vector), the receiver's MLP capacity is also increased from 64 to 1024 in both layers. We present results for the *OO-game different* setup played with $128 \times 128$ Caltech-101 images [10]. The increased number of classes in Caltech-101 may explain the drop in the communication rate in this particular game setting, which compared to the same model played under the original game setup (see the ImageNet-pretrained model in Table 4), is with almost 30% lower. As one might expect, the wider model allows for more details to be captured, and, hence, conveyed in the sketches. Unlike the baseline model which, in this object-oriented setup, develops a communication system that is more representative to the class than to the instance (as discussed in Section 4.3), the wider model starts to draw distinctive representations for objects of the same type. More sketches can be found in Appendix G where one can observe the difference between all images with chairs, for example.

### 4.5 How does the texture/shape bias of the visual system alter communication?

Next, we show that a texture or shape bias of the visual system influences visual communication. This experiment was run under the original game setup with $128 \times 128$ Caltech-101 images [10]. The results shown in Table 4 suggest that inducing a "shape bias" into the model does not significantly improve the agent's performance in playing the game, but produces more meaningful drawings. By using the VGG16 weights pretrained on Stylized-ImageNet [15], the communication protocol also becomes more faithful to the actual shape of the objects. A shape-based sketch is much more interpretable to humans, as it has been known for a long time that shape is the most important cue for human object recognition [27]. Further results from this experiment can be found in Appendix H.

Table 4: **The effect on the communication protocol of using a VGG16 feature extractor network pretrained on datasets that have texture (ImageNet) or shape (Stylized-ImageNet [15]) bias.** Examples are from agents trained using the *original* game with Caltech-101 data. The shape-biased sketches are better at capturing the overall object form, particularly for things like faces.

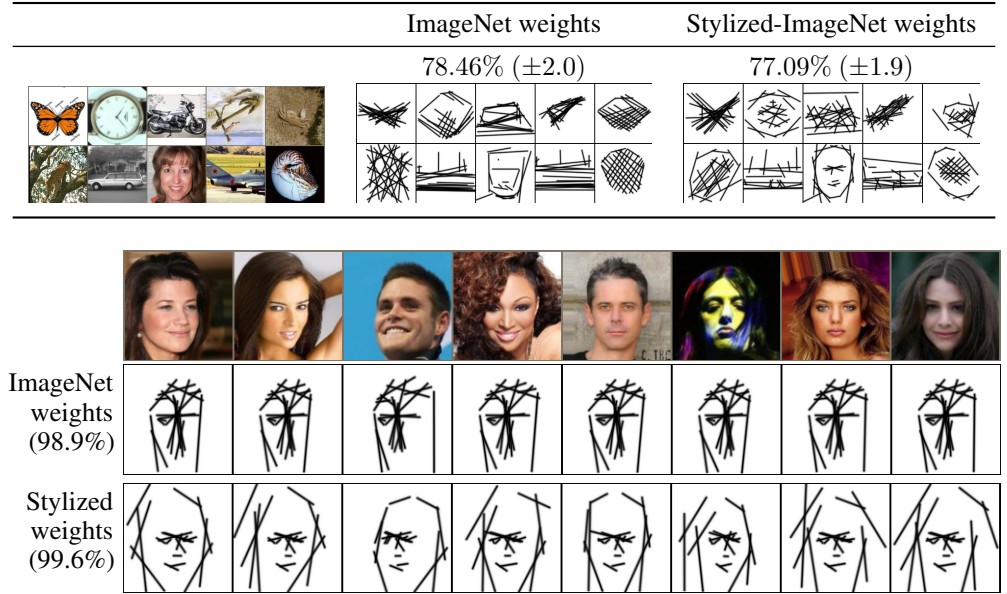

Figure 3: **Sketches from *original* variant games using the CelebA dataset with perceptual loss and different biases from backbone weights.** Both the texture-biased (ImageNet) and shape-biased (Stylized-ImageNet) settings exhibit near-perfect communication success, but the shape-biased sketches are considerably more interpretable and show visual variations correlated with the photos.

## 4.6 Do the models learn to pick out salient features?

From the results we have presented so far, it is evident that, particularly with the perceptual loss, the sender agent is able to broadly draw pictures of particular classes of object. The high communication rates in the *original* game setting would also suggest that the drawings can capture something *specific* about the target images that allow them to be identified amongst the distractors. To further analyse what is being captured by the models we train the agents in the original game setting (using both normal and stylized backbone weights) with images from the CelebA dataset [32], which we take the maximal square centre-crop and resize to 112px. As this dataset contains only images of faces, messages between the agents will have to capture much more subtle information to distinguish the target from the distractors. Results are shown in Figure 3; the communication rate is near perfect for both models, but the difference between the texture-biased and shape-biased models is striking. There is subtle variation in the texture biased model's sketches which broadly seems to capture head pose, but the overall sketch structure is similar. In the shape-biased model head pose is evident, but so are other salient features like hairstyle and (see Appendix I) head-wear and glasses.

## 4.7 Do agents learn to draw in a fashion that humans can interpret?

In order to assess the interpretability of sketches drawn by artificial agents, we set up a pilot study in which a 'sender' agent, pretrained in five different game configurations on STL10, is paired up with a human 'receiver' to play the visual communication game. For this pilot study, we collect results from 6 human participants. Each participant played a total of 150 games, i.e. had to select the target image for each of the 150 sketches drawn by a pretrained sender. Depending on the game setting, the list of options differs, but it is composed of distractors and the true target image. The experimental setup is detailed in Appendix K. Table 5 compares the averaged human gameplay success to that of a trained 'receiver' agent. The results show that the addition of the perceptual loss leads to statistically significant improvement of humans' ability to recognise the identity of sketches. For the original game setting, played in this study with $K = 9$ distractors which might be of the same category as

Table 5: **Human Evaluation results, no learning allowed.** Trained agents communicate successfully between themselves in all settings. Addition of the perceptual loss allows humans to achieve significantly better than random performance (images from STL-10, original games have 9 distractors/game for these experiments & random chance is 10%). In addition, humans are better at guessing the correct image class when the models are trained with the additional perceptual loss.

| Game | Loss | Lines | Agent comm. rate | Human comm. rate | Human class comm. rate |
|---|---|---|---|---|---|
| original | $l = l_{game}$ | 20 | 100% | 8.3% ($\pm$5.4) | 15.0% ($\pm$2.5) |
| original | $l = l_{game} + l_{perceptual}$ | 20 | 93.3% | 38.3% ($\pm$2.5) | 55.6% ($\pm$7.1) |
| original | $l = l_{game} + l_{perceptual}$ | 50 | 100% | 37.2% ($\pm$5.9) | 47.8% ($\pm$7.4) |
| oo diff | $l = l_{game} + l_{perceptual}$ | 20 | 83.3% | 23.9% ($\pm$6.2) | 23.9% ($\pm$6.2) |
| oo diff | $l = l_{game} + l_{perceptual}$ | 50 | 90.0% | 38.9% ($\pm$9.9) | 38.9% ($\pm$9.9) |

the target, we also assess the ability of participants to recognise the class of the sketch. The human class communication rate shows that humans are better at determining the class of the sketch rather than the specific instance, even in the case of sketches generated with the game loss only. In the appendices, we extend the discussion of these results and look into whether communication with an agent can be improved if the human participants are allowed to learn via feedback.

## 5 Conclusions and Future Work

We have demonstrated that it is possible to develop and study an emergent communication system between agents where the communication channel is visual. Further, we have shown that a simple addition to the loss function (that is motivated by biological observations) can be used to produce messages between the agents that are directly interpretable by humans.

The immediate next steps in this line of work are quite clear. It is evident from our experiments that the incorporation of the perceptual loss dramatically helps produce more interpretable images. One big question to explore in the future is to what extent this is influenced by the original training biases of the backbone network — are these drawings produced as a result of the original labels of the ImageNet training data, or are they in some way more generic than that? We plan to address this by exploring what happens if the weights of the backbone are replaced with ones learned through a self-supervised learning approach like Barlow twins [44]. We would also like to explore what happens if the agents' visual systems had independent weights.

Going further, as previously mentioned, learning a perceptual loss would be a good direction to explore, but perhaps this should also be coupled with a top-down attention mechanism based on the latent representation of the input. An open question from doing this would be to ask if this allows for a richer variation in drawing, and for features to be exaggerated as in the case of a caricature. Such an extension could also be coupled with a much richer approach to drawing, with variable numbers of strokes, which are not necessarily constrained to being straight lines. Coupling feedback or attention into the drawing mechanism itself could also prove to be a worthy endeavour.

We hope that this work lays the groundwork for more study in this space. Fundamentally our desire is that it provides the foundations for exploring how different types of drawing and communication — from primitive drawings through to pictograms, to ideograms and ultimately to writing — emerges between artificial agents under differing environmental and internal constraints and pressures. Unlike other work that 'generates' images, we explicitly focus on learning to capture *intent* in our drawings. We recognise however that our work may have broader implications beyond just understanding how communication evolves. Could for example in the future we see a sketching agent replace a trained illustrator? The creation of messages for communication inherently involves elements of individual creative expression and adaption to the emotive environment of both the sender and receiver of the message. Our current models are clearly incapable of this, but such innovations will happen in the future. When they do we need to be prepared for the surrounding ethical debate and discussions about what constitutes 'art'. This has already been seen in the domain of robot art in which Pix18 [31] is a trailblazer as it is not only a robot that paints oil on canvas but can also conceive its own art subject with minimal human intervention.

## Acknowledgments and Disclosure of Funding

D.M. is supported by the EPSRC Doctoral Training Partnership (EP/R513325/1). J.H. received funding from the EPSRC Centre for Spatial Computational Learning (EP/S030069/1). The authors acknowledge the use of the IRIDIS High-Performance Computing Facility, the ECS Alpha Cluster, and associated support services at the University of Southampton in the completion of this work.

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
