# Learning to Draw: Emergent Communication through Sketching
# Appendices

## Contents

## A How does the sketch complexity influence communication?

An interesting question one might ask about a model that learns to communicate by drawing is how complex the sketch image needs to be so that its meaning can be conveyed successfully and communication can be established. We attempt to answer this question by varying the number of lines that our model is allowed to draw to represent the input photograph. In Table I, we show results for experiments run with 5, 10 and 20 lines allowed for sketching. As with previous experiments, we provide the communication success rate with standard deviation over 10 seeds and qualitative results under two game setups. Under the original game format, contrary to what one might expect, the communication rate decreases as the number of lines is increased (see also Table II). From a visual point of view, using more lines results in sketches that are more interpretable to a human observer, although that does not seem to correlate with the agent's communication strategy. Varying the complexity of drawings in the object-oriented game does not significantly influence the communication rate. The sketches, however, show once more that such a setup can induce a more interpretable communication channel. It is clear that even when drawing 5 lines, the model is trying to capture the overall shape of the object.

Table I: **The effect of the drawing complexity (5, 10 or 20 line strokes) on the emergent visual communication channel.** The communication success rate (i.e. receiver agent correctly guessing the target image) and standard deviation across 10 runs are shown next to sample sketches.

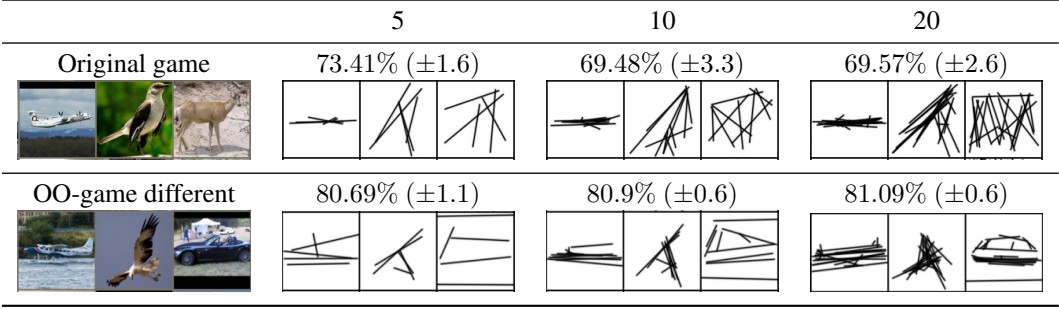

| | 5 | 10 | 20 |
|---|---|---|---|
| Original game | 73.41% (±1.6) | 69.48% (±3.3) | 69.57% (±2.6) |
| OO-game different | 80.69% (±1.1) | 80.9% (±0.6) | 81.09% (±0.6) |

Further, we show results with increased number of strokes, in the original game setting, in Table II. Compared to the model trained to draw with 20 lines in the original game setting (see Table I) which tries to cover the overall space occupied by the photograph's main object, the models trained with more strokes start to draw different lengths, and thus, the object becomes visually more recognisable.

Table II: **The effect of increasing drawing complexity (30, 40 or 50 lines) in the original game setting.** Sketches become visibly more correlated with the input photographs as the increase in the number of line allows for shorter strokes to be used which help with the overall interpretability.

| | 30 | 40 | 50 |
|---|---|---|---|
| Original game | 71.13% (±1.9) | 70.01% (±2.1) | 69.21% (±1.4) |

## B How important is the rasteriser?

To further challenge our hypothesis about visual communication being possible between fully self-supervised agents, we ask the question of how important the rasteriser, and hence the sketch, is for the emergent communication protocol. Instead of line strokes, we constrain the agents to encode images into a cloud of points. We observe that communication between agents is definitely possible even when extracting as little as 10 points from an image, but the resulting sketch does not have any meaning to a human observer. When increasing the number of points to 50, or better 100, the communication success slightly drops to 0.71, 0.66 respectively, but object contours/shapes start to emerge in the sketches as shown in Table III. Encoding to a cloud of points is possible but less

efficient, as it requires more coordinates to be learned to create sketches that are interpretable (to some extent) for humans.

Table III: **The effect of encoding the images into a cloud of points (10, 50, 100) in the original game setting.** Communication is possible with a points rasteriser, but more inefficient. More interpretable sketches require a larger number of points and, hence, more parameters to be learned.

| Points | 10 | 50 | 100 |
|---|---|---|---|
| Original game | 75% | 71% | 66% |

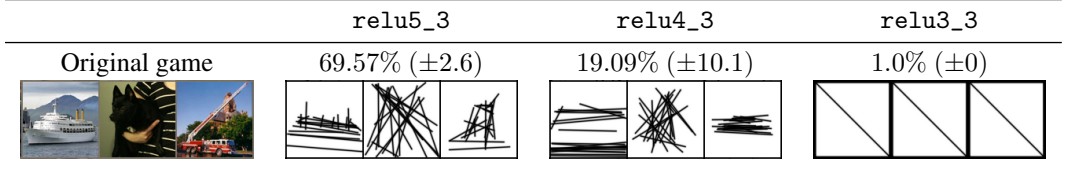

## C What is the impact of L in the computation of the perceptual loss on the emergent sketches?

When computing the additional perceptual loss to induce sketches to become visually more similar to the target photographs, we use the outputs of $L = 5$ feature maps, extracted from the VGG16 layers immediately before the max-pooling layers (`relu1_2`, `relu2_2`, `relu3_3`, `relu4_3` and `relu5_3`), we will refer to this set of feature maps as $fmaps$. Table IV shows the effect of decreasing $L$ and using feature maps only up to the specified layer. More concretely, in the table results for `relu4_3` show how the sketches look like when the perceptual loss is computed over features extracted after `relu1_2`, `relu2_2`, `relu3_3`, `relu4_3` only. We perform this ablation study in the original game setting with 20 line sketches, and show qualitative examples, the communication success rates averaged over 10 seeds and standard deviations. Note that these results are from when $L$ is changed for both sender and receiver agents. We observe that there is a drastic drop in the communication success rate as $L$ decreases from 5 to 4. Even more, if the perceptual loss is computed over the features extracted up to the third block of the VGG16 extraction network (i.e. anything up to `relu3_3`), the model no longer converges and the communication completely fails.

Similarly, Table V shows the effect of increasing $L$. To the original set of feature maps ($fmaps$) used in the computation of the perceptual loss, the output of the other convolutional layers in a certain block (5, 4 or 3) of the VGG16 feature extraction network are added. The results show that increasing the number of feature maps neither impacts the communication success rate nor makes the sketch visually more similar to the corresponding image.

Table IV: **Ablation study on the number of feature maps extracted from the visual system.** Studying the effect of decreasing the number of feature maps (L) extracted from the backbone VGG16 network. We present results by using features extracted from layers in $fmaps$ up to the specified layer.

| | relu5_3 | relu4_3 | relu3_3 |
|---|---|---|---|
| Original game | 69.57% ($\pm$2.6) | 19.09% ($\pm$10.1) | 1.0% ($\pm$0) |

## D What happens if the communication is constrained under an arbitrary, meaningless objective?

One might ask what happens to the communication protocol when the perceptual loss is replaced with some meaningless, arbitrary objective. To explore this scenario, we constrain sketches to look like a single image of a dog (shown in the top left of Figure I) and train agents to draw in order to communicate about CelebA images. As one might expect, the artificial agents can still establish a successful communication strategy about the correct target even when constrained to draw dog-like

Table V: **Ablation study II on the number of feature maps extracted from the visual system.** Studying the effect of **increasing** the number of feature maps (L) extracted from the backbone VGG16 network. To the original set of feature maps, $fmaps$, we add the following extra layers: `reluX_*` indicates that the other feature maps from the $X^{th}$ block of convolutions in the VGG16 feature extraction network are being used to compute the perceptual loss.

| | `relu5_*` | `relu5_*, relu4_*` | `relu5_*, relu4_*, relu3_*` |
|---|---|---|---|
| Original game | 68.4% ($\pm$2.0) | 69.5% ($\pm$1.8) | 69.0% ($\pm$1.2) |

sketches. Figure I show results for models trained with such an additional objective, fully or partially, by scaling $\lambda$ in $l = l_{game} + \lambda l_{arbitrary}$. These results show that it matters what the perceptual loss is: if it constrains sketches to look like the corresponding photographs, a human receiver might have a chance at recognising the person, but with such an arbitrary objective, humans stand no chance at understanding which image the sender agent tries to communicate about. Agents' communication success rate is also impacted (compared to the model with Stylized weights trained with our $l_{perceptual}$ and $\lambda = 1$, results shown in Figure 3 of the paper).

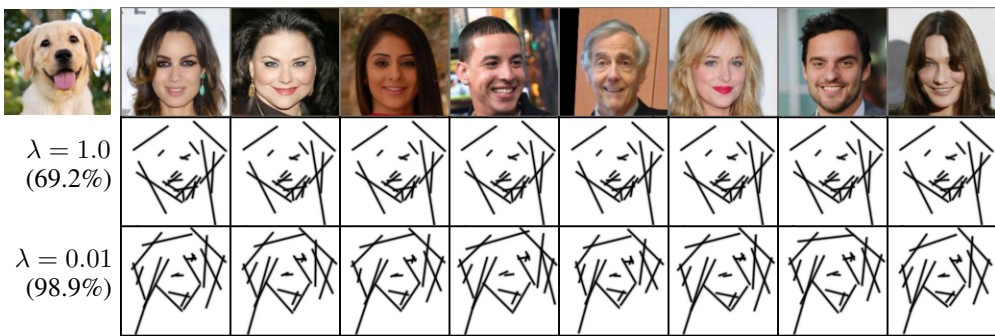

Figure I: **Sketches from *original* variant game using the CelebA dataset with an arbitrary objective: the sketches are constrained to look like the image of a dog (fully or partially, by scaling the perceptual loss coefficient $\lambda$).** Results are shown for a model with the visual extraction network pretrained on Stylized-ImageNet.

# E   What happens when injecting Out-of-Distribution images?

To further investigate the emergent visual communication protocol, we test a pair of agents pretrained in the proposed framework on out-of-distribution images. More specifically, we evaluate a pair of agents, previously trained in the *original* setting on CelebA dataset, on games played with STL-10 images. Agents with ImageNet-pretrained visual systems, achieve a test communication rate on STL-10 of $15.8\%$. Similarly, agents initialised with Stylized-ImageNet weights achieve $30\%$ test recognition accuracy. It is worth noting that even if these results are significantly lower, they are still better than random chance, particularly with the stylized imagenet weights, where the sketches have considerably more diversity (but still all look like faces rather than the objects in the images).

A similar experiment is performed with models pretrained on STL-10, with either just the $l_{game}$ or with the additional $l_{perceptual}$. When testing these on Caltech-101 test data, the communication success drops to $22.2\%$ and $26.7\%$ respectively. It is interesting that the perceptual loss helps improve generalisability in this case.

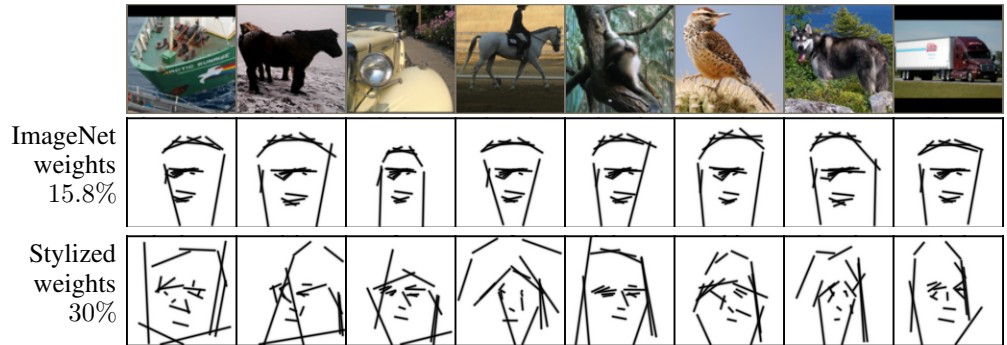

Figure II: **Sketching agents, previously trained on CelebA (original game) tested on STL-10 test images.** We compare models with visual systems pretrained on ImageNet and Stylized-ImageNet.

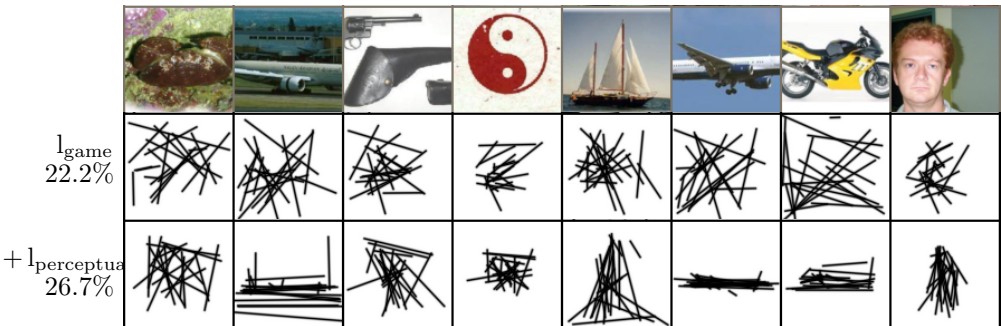

Figure III: **Sketching agents, previously trained on STL-10 (original game) tested on Caltech-101 test set.** We compare models pretrained with $l_{game}$ only with those that also use $l_{perceptual}$.

## F  Sketching under different game setups

Table VI illustrates more examples of sketches drawn under different game configurations, as discussed in Section 4.3 of the paper. Clearly, some classes are better represented and more interpretable to a human than others. Overall, the object-oriented game setups, especially *OO-game different*, push the visual communication channel towards a more class-specific representation.

Further, in Figure IV, we provide an example of the full reference games to help the reader understand how difficult the original game setting, with 99 distractors, would be to play for a human receiver. This should shed some light on how "interpretable" the communication is in the full context given to the receiver agent, which may contain many perceptually similar distractors in the original setting. The game in either of the object-oriented settings shown in Figures V and VI, played on STL10 classes, seems much more feasible to a human receiver.

Sender image:

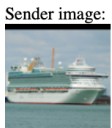

Sketch:

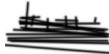

Receiver images:

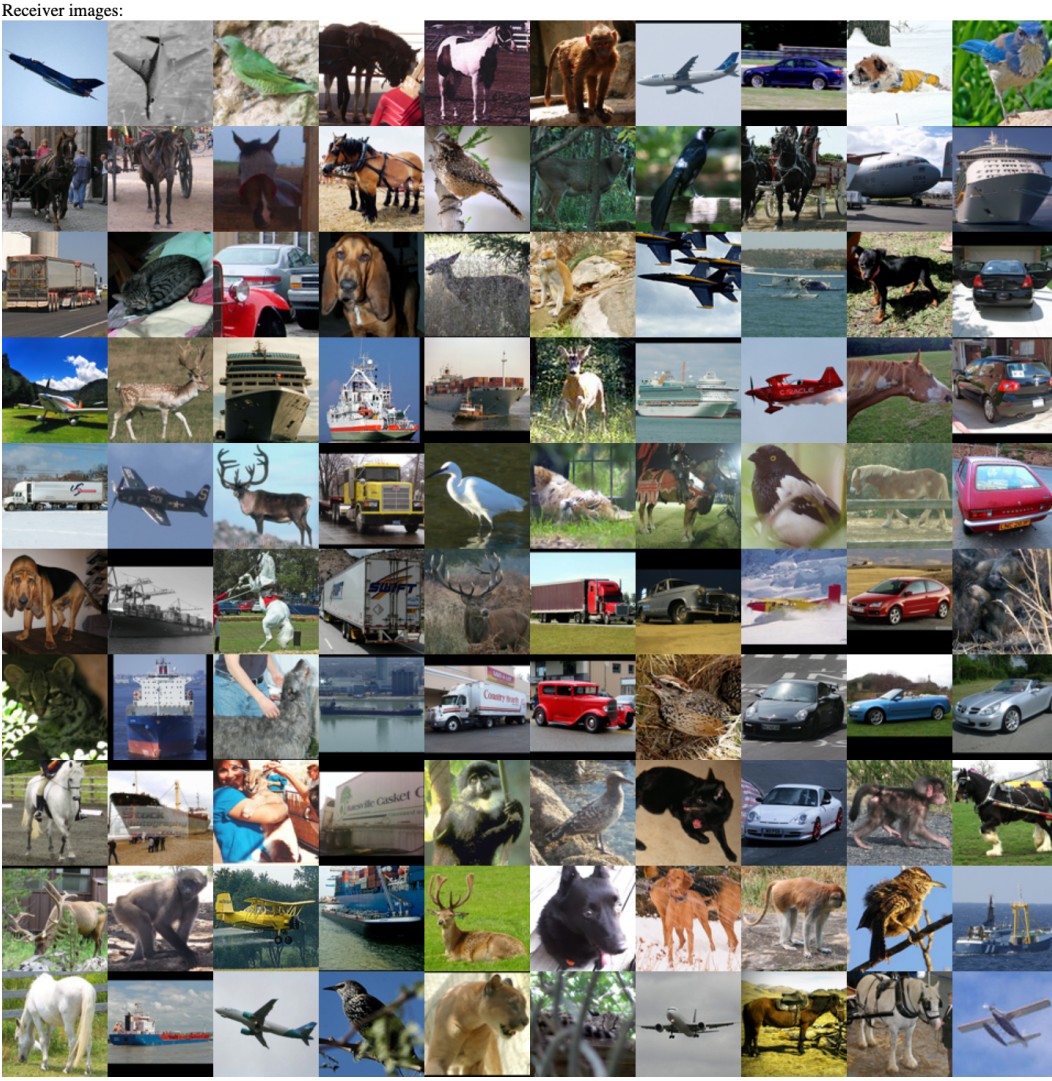

Figure IV: **Example of full reference game -** *original* **setting with 99 distractors.**

Sender image:

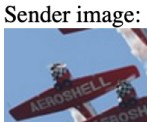

Sketch:

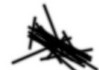

Receiver images:

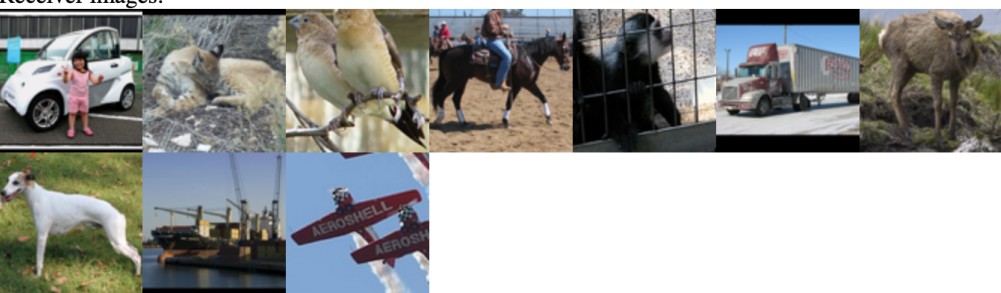

Figure V: **Example of full reference game - *object-oriented same* setting** in which the sender's target is part of the set of images shown to the receiver.

Sender image:

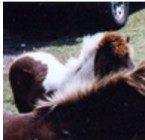

Sketch:

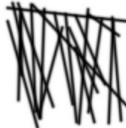

Receiver images:

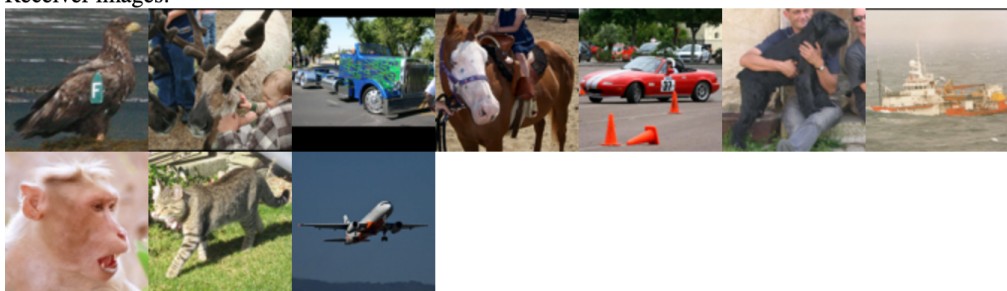

Figure VI: **Example of full reference game - *object-oriented different* setting** in which the receiver's target is a different photograph that belongs to the same class as the sender's image.

Table VI: **More example sketches produced by the agents in the three different game setups using the** $l_{game} + l_{perceptual}$ **loss.** Examples are from STL-10.

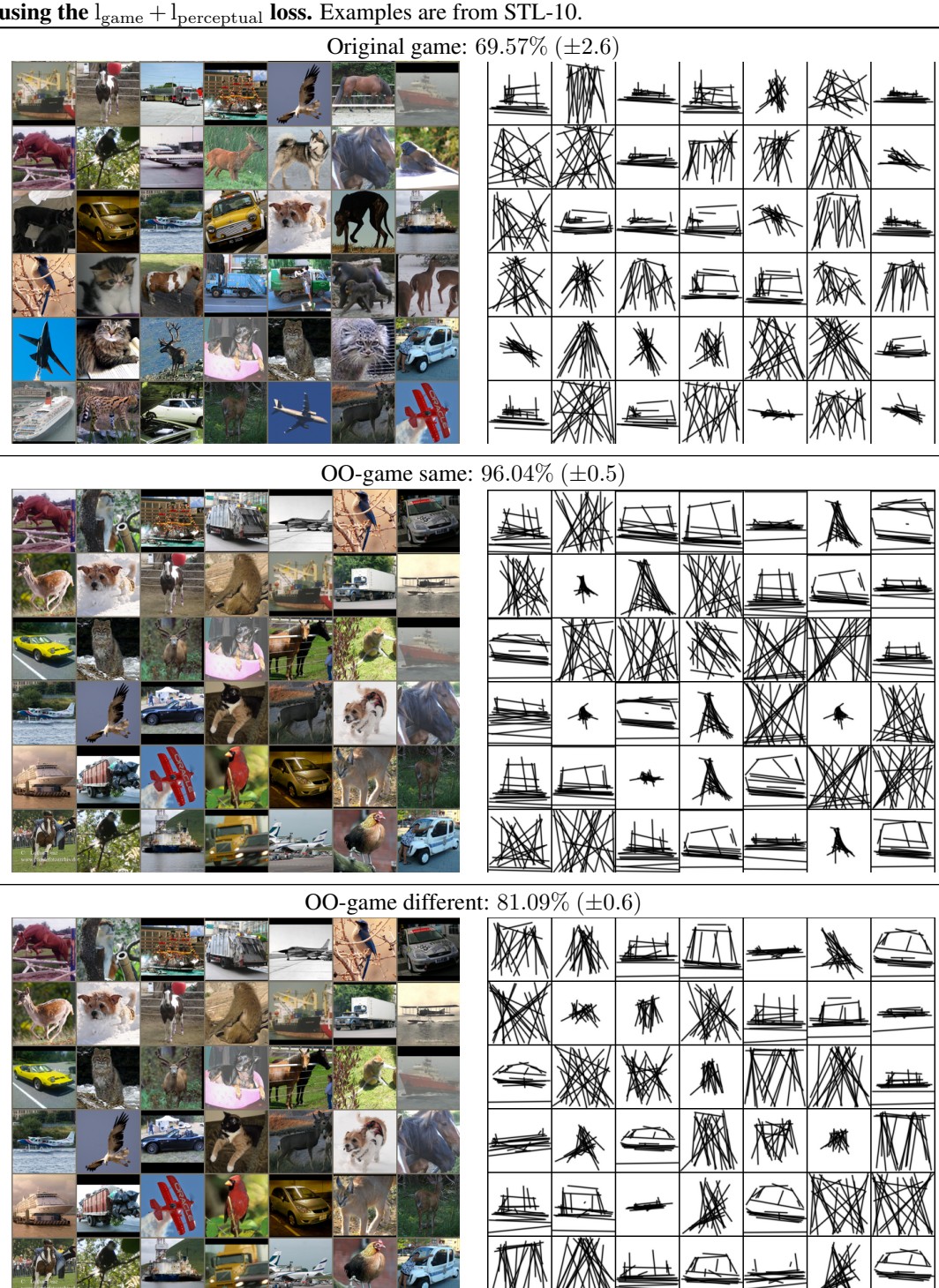

Original game: 69.57% (±2.6)

OO-game same: 96.04% (±0.5)

OO-game different: 81.09% (±0.6)

## G Varying model capacity on Caltech-101

In Table VII we provide additional examples to illustrate the effect of the model's capacity on the emergent sketches. More instance-specific details are captured by the wide model although the game is played in the *OO-game different setting*.

Table VII: **The effect of the model's capacity on its sketches.** Examples from training on $128 \times 128$ pixel Caltech-101 images, in the *OO-game different* setting.

# H    The effect of pretraining the visual system with texture/shape bias

Table VIII shows further examples of the influence the texture/shape bias has on the drawing quality.

Table VIII: **The effect of pretraining the VGG16 feature extractor network with a texture (ImageNet) or shape (Stylized-ImageNet) bias.** Examples are from agents trained in the *original* game setup with $128 \times 128$ Caltech-101 images. Shape-biased sketches are, visually, more similar to the objects they represent.

# I  Do the models learn to pick out salient features?

Figure VII provides additional results of the experiment discussed in Section 4.6, which looks at the ability of the model, either texture or shape-biased, to capture salient features. It is clear that the shape-biased sketches are visually more correlated with the photos.

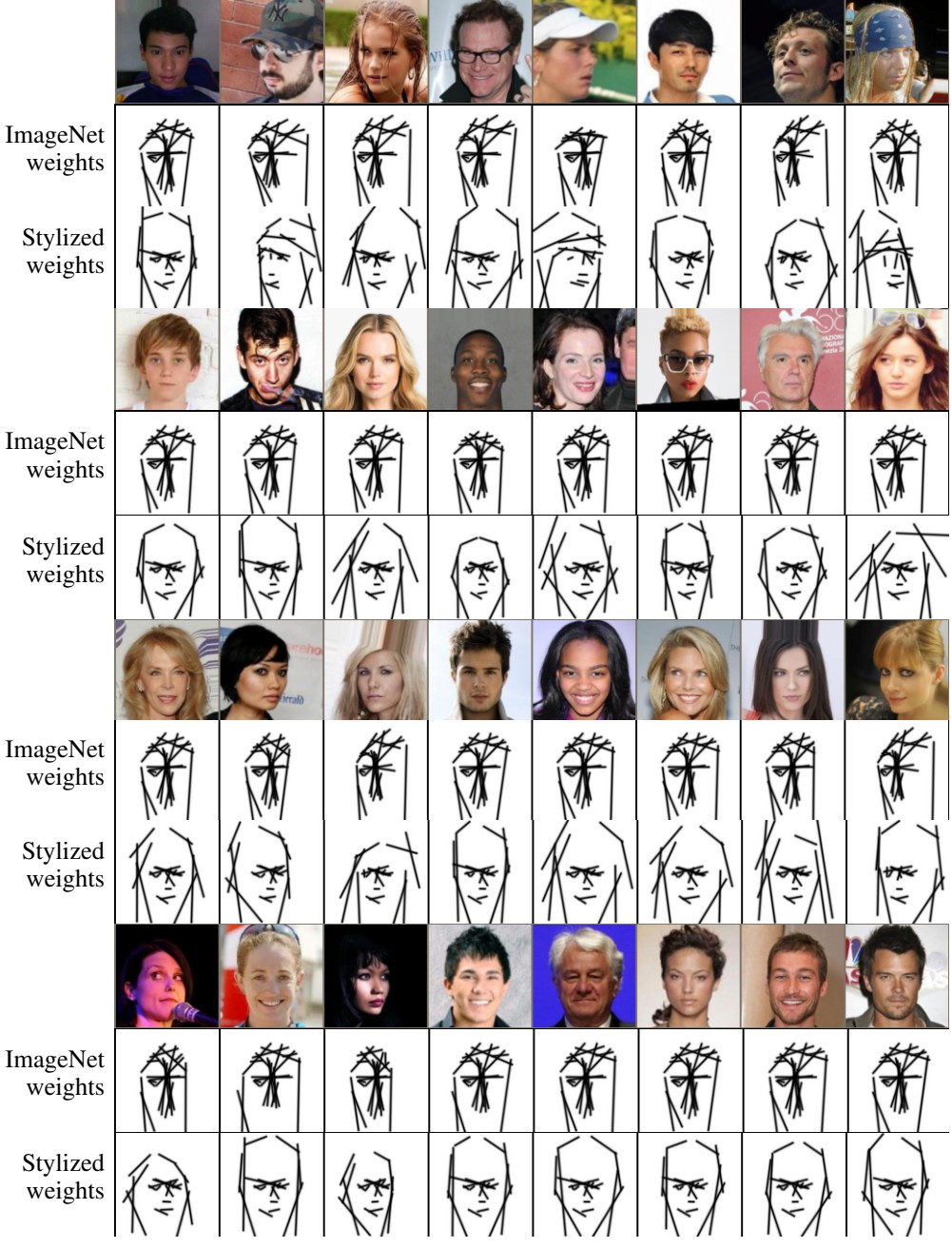

Figure VII: Sketches from *original* variant game using the CelebA dataset, the perceptual loss and different biases from backbone weights. Although both models have near perfect communication success, it is clear the inducing a shape bias helps bring out the most salient and distinctive features.

## J How much do sketches differ visually across seeds?

Throughout the paper, the sample sketches are presented from one seed out of the 10 model runs. Here we include an example of an overlay of the 10 seeds, normalised to look like a heatmap so that darker lines represent strokes generated by *more* models. As can be seen in Figure VIII, the 10 different models trained on Caltech 101 from different seeds are consistent in picking out key features of the input image, but have variation in finer details.

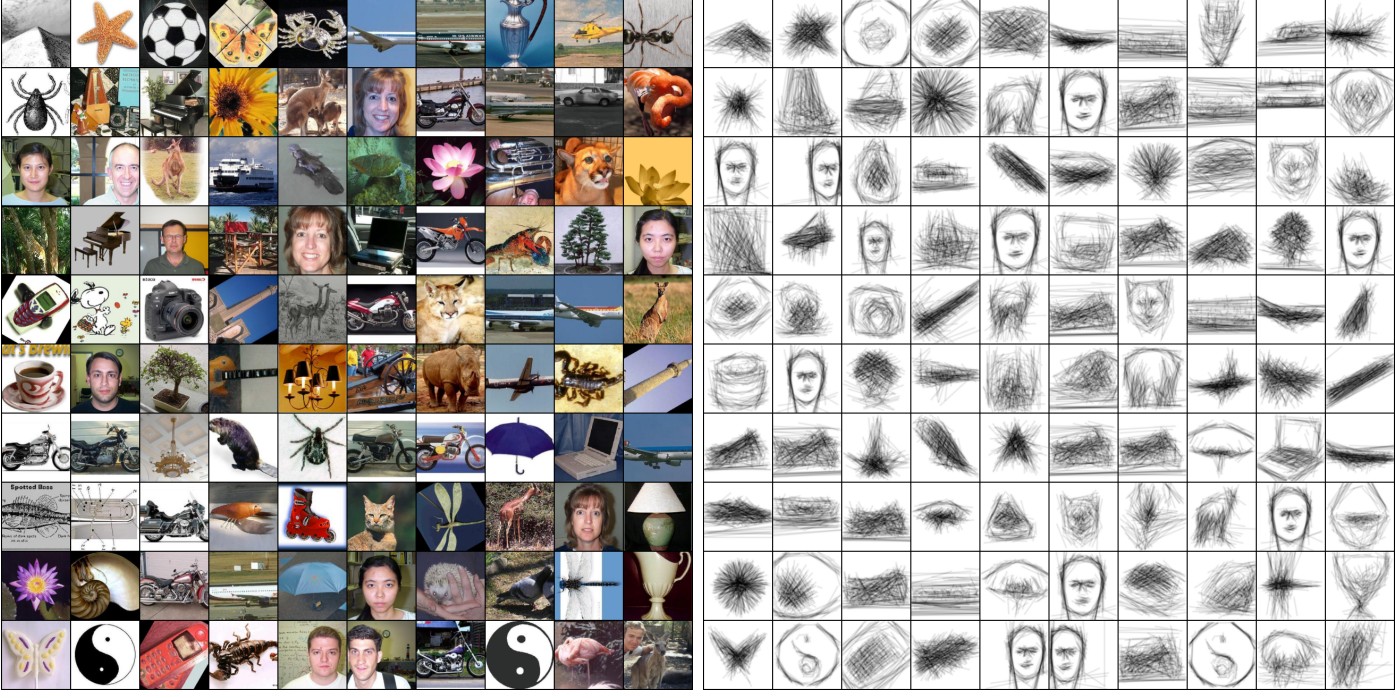

Figure VIII: **An overlay of 10-seeds sketches** drawn by a model trained in the *original* game variant on Caltech101, with Stylized-ImageNet weights.

## K  Human Evaluation Experiment Details

This section details the setup and results of our human evaluation experiment. As mentioned in Section 4.7, we perform a pilot study that looks at gameplay success when the receiver agent role is played by a human participant.

### K.1  Experimental Setup

**The task**  To reiterate the experimental task, the human participant is shown a sketch (previously generated by a trained Sender agent during model evaluation) and is asked to select by clicking the corresponding target image from a grid of images, as illustrated in Figure IX.

**The data**  The sketches used in this experiment are generated in five different game configurations, varying game setup, agents' training objective and the number of strokes. For the purpose of this study, the sketches are generated by models trained with the same fixed random seed. Whilst there is inevitably some variation in models from different seeds (see Appendix J), this is not explored in the human evaluation.

For each game setting, the participant played 30 games, matching a total of 30 sketches to different target image sets. Each human participant played a total of 150 games, and the total amount of data collected in the pilot study corresponds to 1800 games. For the purpose of this study, the games were chosen randomly from all those possible within the STL-10 test dataset. For all game settings used in the human pilot study, we limit the number of distractors to $K = 9$.

**User interface**  To allow human participants to play the game, a web interface was developed and each participant was provided with a set of 5 unique URLs corresponding to the 5 different game settings. Information on what the different settings involved was not provided to the participants. Each URL took the participant through 30 games and stored their answers in a database.

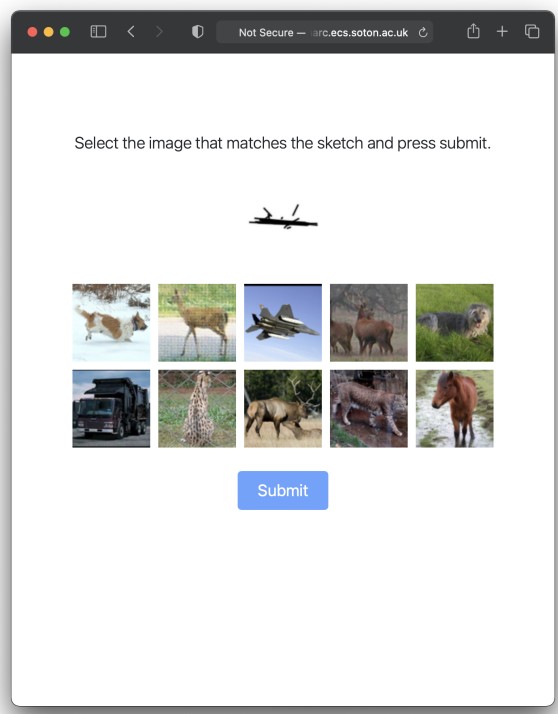

Figure IX: **Example of a game, original setting - the human participant has to pick from 10 images.**

An example of such a game is shown in Figure IX. We do not impose a time limit per game, but record how much time the participants take to make their guess. Figure X shows our admin interface which summarises the averaged statistics based on the games played in this pilot study. Figure XI shows the interface when feedback is given (see Appendix K.2).

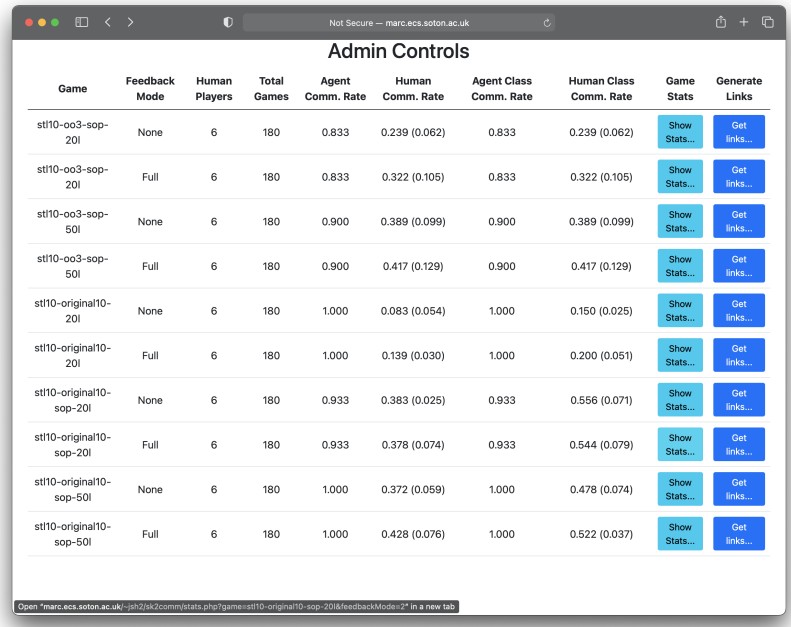

Figure X: **The admin interface.**

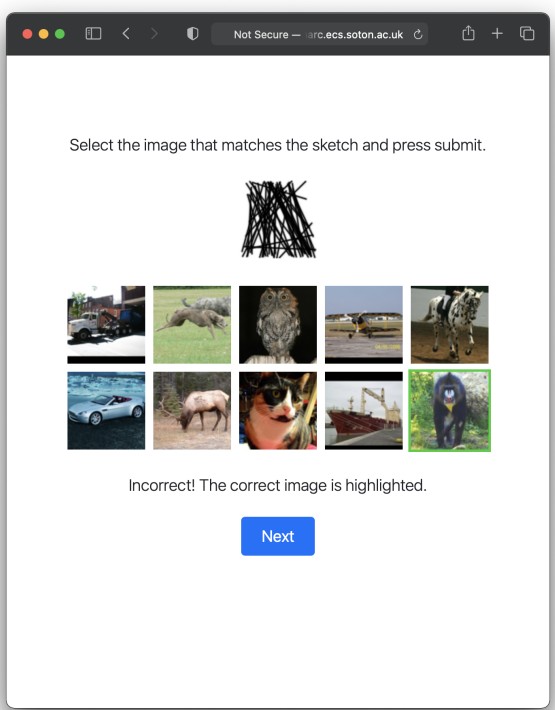

Figure XI: **Example of the game played with feedback.**

**Participants** We divide the human evaluation into two disjoint study groups: participants who just play the game with no feedback and, hence, cannot learn during game-play (results are presented in Table 5), and a second group which is allowed to learn from feedback. Details about the latter group are discussed in Appendix K.2.

For the purpose of the study, we collect results from 6 participants per group. Overall, the study includes participants aged between 20 to 35 with various professions. Participation in the study does not require any specific skills.

## K.2 Can human participants *learn* to play the game?

The principal pilot study (Section 4.7) is looking at humans' ability to play the game with an agent, with no feedback involved. The human participants will not know what the correct target was or if they guessed correctly. We also pose a slightly different question: Can humans learn to play the game with an agent? For this secondary study, after participants select what they believe to be the target image, they will be told if their selection was correct or not and the correct target will be indicated (as shown in Figure XI).

Table IX summarises the statistics computed over the participants in this secondary study. The participants were tested on the same set of games as the first group, and the same metrics are reported.

T-tests run between the averaged communication success rates of the same game setting in the group with feedback versus the one without feedback, do not show a statistically significant improvement when participants are allowed to learn from feedback, except for the original game with $l = l_{game}$ only. As we expected, participants in both study groups had the lowest scores in this game across all tested settings: without feedback the averaged $commrate = 8.3\%(\pm 5.4)$; with feedback, $commrate = 13.9\%(\pm 3.0)$. The sketches drawn by a Sender agent pretrained without the perceptual loss are not "constrained" to resemble the target image, hence they are the least interpretable. However, the two-tailed P-value between the two groups performance in this setting was less than 0.0001 which suggests that feedback can lead to a statistically significant improvement when the sketches are not visually interpretable. Still, this is by far the worst communication scenario. This is also indicated by the amount of time the participants spent on average on this game which is higher than in other settings, the majority taking between 1 minute and 2 minutes 30 seconds per sketch.

In the future, it would perhaps be interesting to explore if humans could learn with feedback if they were to play more games; the 30 games per setting used in this experiment is possibly too little to allow a human player to robustly learn the strategy used by the agent.

## K.3 Extended discussion of results

**Does the addition of the perceptual loss give statistically significant improvement over games which use only the hinge loss?** All participants were asked to play the original game with 20 stroke-sketches produced when $l = l_{game}$ and also when $l = l_{game} + l_{perceptual}$. Performing t-tests between the averaged communication rates within each study group, with and without perceptual loss, resulted in P values less than 0.0001, which indicates that the perceptual loss leads to a statistically significant improvement in humans' ability to play the game with the agent.

**Does the number of strokes influence human performance?** We tested the original game and the object-oriented game setup, each with 20 and 50 strokes. The results indicate that in both settings,

Table IX: **Human Evaluation results, learning allowed from feedback.**

| Game | Loss | Lines | Agent comm. rate | Human comm. rate | Human class comm. rate |
|------|------|-------|------------------|------------------|------------------------|
| original | $l = l_{game}$ | 20 | 100% | 13.9% ($\pm 3.0$) | 20.0% ($\pm 5.1$) |
| original | $l = l_{game} + l_{perceptual}$ | 20 | 93.3% | 37.8% ($\pm 7.4$) | 54.4% ($\pm 7.9$) |
| original | $l = l_{game} + l_{perceptual}$ | 50 | 100% | 42.8% ($\pm 7.6$) | 52.2% ($\pm 3.7$) |
| oo diff | $l = l_{game} + l_{perceptual}$ | 20 | 83.3% | 32.2% ($\pm 10.5$) | 32.2% ($\pm 10.5$) |
| oo diff | $l = l_{game} + l_{perceptual}$ | 50 | 90.0% | 41.7% ($\pm 12.9$) | 41.7% ($\pm 12.9$) |

a higher number of strokes leads to better communication. However, in the group without feedback (Table 5), the mean communication rate was similar for the original setting with 20 strokes and with 50 strokes. The same game setting tested by people with feedback, however, showed a small increase in overall communication success. One should take into account that in this game setting, the human player might have to choose between more images from the same class. For an artificial agent, this can be an easy task. However, we might envisage a scenario in which other characteristics of a drawing would be included, such as colour, which might help the human differentiate between multiple instances from the same class. For example, think of 3 different species of birds, which could all be represented by some very general sketch, but could become distinctive if the colour were to be included. In the object-oriented game setting, the gap between 20-stroke and 50-stroke games is a bit more significant for both study groups.

**Are humans better at determining the broader class of a sketch than at recognising the specific instance?**    In the original game setting, it is possible to encounter distractor images from the same class as the target. In addition to the communication rate measure, which shows the overall success of an agent (human in this case) selecting the correct target image, we also compute the class communication rate, which calculates the overall success of an agent selecting an image from the same class as the true target. T-tests run between human communication rate and human class communication rate in the original game settings showed a statistically significant difference in both study groups. Humans are significantly better at understanding the broad class than they are at determining a specific instance based on the sketch in the games where there are multiple targets of the same class. This effect is possibly weakened by an increase in the number of strokes, however, as evidenced by a consistent lowering of statistical significance.