# OpenReview forum: "Learning to Draw: Emergent Communication through Sketching"
_NeurIPS.cc/2021/Conference — NeurIPS 2021 Oral_

### Official Review · Reviewer_9t7y · 2021-07-08

**Rating:** 7
**Confidence:** 4

**Summary:**

I’m impressed by the authors’ inclusion of a human evaluation. While they paint a somewhat unclear story about human interpretability (hard to tell how good 38% is), including the results in commendable and these kinds of studies are sorely missing from the literature.

I’ve increased my score accordingly. I’m borderline on increasing my score even further: I think I would have liked to see some of the other experiments I suggested in the initial review to really nail down the interpretability idea more completely. But still, this is a very solid paper now—kudos to the authors!

---

This paper is a study of emergent communication in an image-based referential game where the communication channel between two agents is not discrete sequences of tokens, but instead *sketches* of images; in other words, the agents learn to communicate via drawing. The speaker uses a recently-proposed differentiable drawing procedure, where line segments generated by an agent are combined and converted to a grayscale image via a relaxed rasterization process. The listener then receives the rasterized drawing and needs to identify the speaker's image from a set of distractor images. The entire procedure is end-to-end differentiable.

Optimizing solely communication success unsurprisingly results in non-interpretable sketching. To encourage the communication to be more interpretable, authors propose adding an auxiliary perceptual loss, where the speaker's drawings are optimized not just for listener success, but also trained to be perceptually similar to the target image, by minimizing distance between their representations according to the speaker's trained vision module. This results in sketches that often capture the rough details of the target image, with little-to-no degradation in communication success.

The authors try several additional experiments, e.g. varying the width of the communication channel, using different pretrained backbones that are more biased towards shape, etc. Many of these results do seem to have some effect, but the comparisons between models are limited to a qualitative comparison between 5 example images, so the conclusions we can gain from these experiments are a bit limited.

**Limitations And Societal Impact:**

Yes, there's a good discussion of future downstream impacts of this work in the discussion.

**Main Review:**

## Strengths

Overall this is a well-written paper with an interesting and extensible idea at its core; modeling sketch-based communication has several potential downstream applications for multi-agent systems and should be of interest to those interested in simulations of the development of communication systems.

## Weaknesses

My main concern, which makes me overall on the fence with this paper, is that the claim that the agents learn to draw "in a fashion humans can interpret" is a very subtle one and may not be adequately discussed or supported in the paper.

One consensus of the existing literature is that agents often learn to exploit inscrutable, unintuitive patterns in the either the input or the communication signal; this is doubly true when the communication protocol is continuous (as is the case here). For example, a small shift in a single pixel value in a speaker's drawing could drastically change the listener's prediction.

The perceptual loss constrains the drawings to "look like" the target image, but this doesn't guarantee that the agents interpret the drawings as humans would. For example, the agents could be ignore the "gestalt" of the sketch, and instead learn that differences in the mean pixel density of some region of the sketch indicate the mean RGB value of some region of the target image. Such degenerate schemes might be successfully learned even under the constraints of the perceptual loss.

One suspicion I have is that communication accuracy would remain successful even if the perceptual loss was some meaningless, arbitrary objective: for example, constrain all sketches to look like a single image (of a dog, perhaps). Since the communication is continuous, and agents are able to pick up on small deviations in the input, they may still be able to play this referential game with a wide variety of images.

Even ignoring these questions, it's insufficient to examine solely the target image and the corresponding sketch in isolation, see some broad similarities, and then declare the communication "human interpretable." A more pragmatic claim of "human interpretability" would show that a human could successfully play the role of the listener and select the right image based on the speaker's sketches. There is no such evaluation in the paper; at the very least it would be good to see some examples of the full reference games (w/ 99 distractors), so we can see ourselves how "human interpretable" the communication is *in the full context given to the listener* (which may contain many perceptually similar distractors, etc). Take Figure 3 for example: even with the stylized image weights, for many of the sketches it's doubtful that humans would be able to correctly identify the target among all images in this figure—even if there are some high-level correspondences between images and sketches. If the agents are able to construct drawings that succeed in reference games where humans would completely fail, is the communication really human-interpretable?

To be fair, these are tough, somewhat philosophical questions. But I would encourage the authors to add more concrete discussion of what it precisely means to be "human interpretable". Finally, to summarize some concrete additions to the paper that would shed light on this ideas:

1. Does it matter what the perceptual loss is? What if we constrain the images under some arbitrary, meaningless objective?
2. Human evaluation - are sketches "interpretable" to a human insofar as a human can play the reference game with the images? Alternatively, consider whether a human could generate sketches for a trained listener?
3. To what extent does the learned communication protocol generalize to other speakers and listeners? If a speaker can play the game with a new listener, this would support the idea that speaker-listener pairs are not devising their own specific and uninterpretable drawing schemes, but instead converging on some shared communication that indeed focuses on learning a general, human-like interpretation of generated sketches.

## Other questions

- How different is OO-game-same from the original setting where distractors are randomly sampled? It seems like most distractors are probably already in separate categories in the first place?

## Missing citations

Some neural models of sketching have been developed in the cognitive science literature, e.g.:
"Pragmatic inference and Visual Abstraction Enable Contextual Flexibility during Visual Communication." Judith Fan, Robert X.D. Hawkins, Mike Wu, Noah Goodman. COBB, 2019

## Minor

- L44 "Maximize gameplay alone" → "maximize gameplay success/communication success?"

**Time Spent Reviewing:**

3

---

> ### Author Response · Authors · 2021-08-10
> **Response to Reviewer 9t7y**
>
> We would like to begin by thanking the reviewer for their time and for the suggestions to help strengthen our work. We respond to specific comments below:
>
>
> > My main concern, which makes me overall on the fence with this paper, is that the claim that the agents learn to draw "in a fashion humans can interpret" is a very subtle one and may not be adequately discussed or supported in the paper.
>
> Sure, this is definitely an area where we can strengthen the argument in the paper by adding much more discussion (although we disagree that this isn't supported in the review version, we agree it could be better supported). In light of the comments from the other reviewers we're currently working on adding a human evaluation (see description in the general comment to all reviewers) that should address this point concretely.
>
> Beyond this, however, we would like to politely point out to the reviewer that the contributions of our paper go well beyond this one aspect, and as noted by all the reviewers opens a doorway to exploring different types of communication systems.
>
> > One consensus of the existing literature is that agents often learn to exploit...
>
> This is undoubtedly true and is precisely the reason that one would somehow want to bias the model (e.g. pretrained with stylized images) and regularise it (e.g. the perceptual loss).
>
>
> > The perceptual loss constrains the drawings to "look like" the target image, but this doesn't guarantee that the agents interpret the drawings as humans would...
>
> We agree that the model might learn to perceive in a different way to a human (there are clearly many ways information might be "leaked" through continuous parameters, and our CNNs are only very crude proxies of the human visual system). We don't however believe that this should be considered to be a problem; if a human (ideally with little or no specific training) can play the communication game successfully, then the actual underlying mechanism (the "features" on which the agents rely) are less important.
>
>
> > One suspicion I have is that communication accuracy would remain successful...
>
> We don't disagree with this (in a different, but related, way this is exactly what Table 2 shows). Equally, we'd probably expect that a pair of human agents might also be able to play these games by drawing such that the sketches are "in the style of a dog". Overall, in the paper, the key point we were trying to make is that with appropriate objectives (the perceptual loss being just one example), it is possible to regularise the artificial agents in such a way that they outwardly behave in a much more human interpretable way (irrespective of their internal representations/perception).
>
>
> > Even ignoring these questions, ... is the communication really human-interpretable? To be fair, these are tough, somewhat philosophical questions. But I would encourage the authors to add more concrete discussion of what it precisely means to be "human interpretable".
>
> This is a fair criticism; as mentioned we intend to address this by adding a human evaluation of gameplay (human receiver agent), and can of course add graphical examples of complete games. As clearly these are hard questions to fully answer, we also intend to expand the discussion of the current limitations to ensure that these concerns are adequately addressed.
>
>
> >Does it matter what the perceptual loss is? What if we constrain the images under some arbitrary, meaningless objective?
>
> We suspect not (in terms of the agents’ communication success rate), but we'll try and actually run the experiment (this is a great addition to reviewer pCdM's comment E on counter-factual experiments).
>
> > Human evaluation - are sketches "interpretable" to a human insofar as a human can play the reference game with the images? Alternatively, consider whether a human could generate sketches for a trained listener?
>
> We are focussing on the former at present (see general comment to all reviewers). We have an idea of how one might do the latter, but it would likely constitute a stand-alone piece of work in its own right.
>
> > To what extent does the learned communication protocol generalize to other speakers and listeners? If a speaker can play the game with a new listener, this would support the idea that speaker-listener pairs are not devising their own specific and uninterpretable drawing schemes, but instead converging on some shared communication that indeed focuses on learning a general, human-like interpretation of generated sketches.
>
> Sure, we will add this extension in the appendices. _Preliminary_ results with two pairs of agents trained on disjoint subsets of STL-10, constrained to draw 20 lines in the original game setup (with 99 distractor images), achieve communication success rates of 72% and 62% respectively. Pairing the Sender of the first model with the Receiver of the second model, and vice-versa results in a much lower $commrate$ of only ~20%. For a similar experiment using the OO-game-different setting, the two individually trained pairs of agents achieve 90% and 85% communication success. When swapping the agents, the new pairs achieve about 60% $commrate$.
>
> We will continue to dig into these results; currently, the implication is that for the original game the agents don't generalise well (but are considerably better than random). The OO-game also shows some generalisation, but it is lower than what we had anticipated. One particular thing we want to check next is that we haven't caused the models to overfit by training on only half the amount of data.
>
> > How different is OO-game-same from the original setting where distractors are randomly sampled? It seems like most distractors are probably already in separate categories in the first place?
>
> This entirely depends on the dataset; for datasets with 10 classes (like STL-10 for example), in the original game (99 distractors) there will be ~10 images of each of the 10 classes presented to the receiver. This inherently means that the model cannot communicate successfully by just sketching a representation of the class alone.
>
> > Missing citations
>
> Thank you for this; we will add it to the discussion on related work.
>
>
> We tried to address all the reviewer's mentioned concerns. If any part of our response is unsatisfactory, we would politely ask the reviewer to specifically point out anything that is still unclear and we will try to provide a more appropriate answer. Finally, we thank them again for their time and suggestions!

---

> > ### Comment · Reviewer_9t7y · 2021-08-10
> > **Response**
> >
> > Thanks to authors for the detailed response to my review! The additional experiments e.g. on generalizing to new populations is greatly appreciated. On the topic of interpretability:
> >
> > > We don't however believe that this should be considered to be a problem; if a human (ideally with little or no specific training) can play the communication game successfully, then the actual underlying mechanism (the "features" on which the agents rely) are less important.
> >
> > Yes, I think this is the key issue—authors are working on some preliminary versions of these experiments but we don't really know the communication is human-interpretable unless a human successfully interprets it.
> >
> > > Equally, we'd probably expect that a pair of human agents might also be able to play these games by drawing such that the sketches are "in the style of a dog"
> >
> > This one is also tricky—not sure how good I would be at communicating the identity of one person among 10 others if I had to draw them as a dog :)
> >
> > Overall, after reading the other reviews and responses, my sentiment remains the same: I believe the claim that communication is truly "human-interpretable" is most convincingly supported by actual human-in-the-loop experiments. That said, I don't think this is necessarily a *fatal* issue, since human-in-the-loop experiments are rare in the literature. The authors promises of (1) qualifying these claims in writing and (2) including some preliminary human experiments alleviates my worries. I will keep my score the same; this is a solid paper and I wouldn't object to seeing it at the conference. If human experiments are indeed posted before discussion period ends I will consider raising my score.

---

### Official Review · Reviewer_pCdM · 2021-07-16

**Rating:** 8
**Confidence:** 5

**Summary:**

This paper is casting the symbolic communication Lewis game into a drawing communication Lewis game.
To do so, they use a VGG network followed by a MLP/rastering function for drawing.
They show that, with a proper loss, the speaker agent may successfully communicate concepts with basic drawing.

While this idea is definitely not novel [1], and it has been in the air again recently [2], the quality of this work is pretty high. It was both an enjoyable and insightful reading. There is a clear effort to mix quantitative and qualitative results (with sketches). Besides, the authors did not stop their research after a few positive results, and have excellent additional experiences in section 4. Nonetheless, the paper still has some minor weaknesses that I am happy to discuss further (cf Main results).

[1] Bartlett, Frederic Charles, and Frederic C. Bartlett. Remembering: A study in experimental and social psychology. Cambridge University Press, 1932.
[2] Fernando, Chrisantha, et al. "From Language Games to Drawing Games." arXiv preprint arXiv:2010.02820 (2020).  to-cite!

**Limitations And Societal Impact:**

cf above.

**Main Review:**

Key remarks:

A) The perceptual loss is one of the key components of the paper. However, I am wondering what is the impact of $L$ on the training. If $L$ is greater/lower, does it reduce accuracy? Interpretability? More generally, I am missing a comprehensive ablation on the paper core elements num_strokes (20+), $L$, $\lambda$ where $l = l_{game} + \lambda l_{perceptual}$ in the appendix would make the current conclusion stronger, and help reproducibilty. Besides, in light on Tab1 Appendix A, the task accuracy seems quite noisy. Std are definitly missing in the paper. How many seeds did you use? Furthermore, having the overlay of num_seeds=10 sketches would be excellent!

B) Regarding the perceptual loss, I would recommend adding a small paragraph to link it to other work that explores low/high feature levels, e.g., [1] or [2] or, more generally, to style transfer [3]. One may argue that the loss is a naive implementation of style transfer.

C) What are the intuitions of using a VGG network instead of ResNet? Using Hinge-loss instead of InfoNCE loss? cf. Tab2 in [4]? Funnily, the paper is using Deep Learning techniques from 2015. Note that it does not undermine the quality of the paper.

D) As one of the major claims of the paper is communication with humans, I would have significantly increased my score to have a human evaluation (i.e. the score of humans playing this game).

E) As mentioned, the paper has many complementary experiments. Yet, I think it is also missing what I call counter-factual experiments, i.e., experiments that try to break the setting to challenge the paper hypothesis further. Those are potential ideas:
 - how important is the Rasteriser (i.e. sketch) -> You can simply extract a cloud of points (which still encode an image)
 - what happens when you inject Out-of-Distribution images?  In Figure 3, put animals instead of human faces? Or similar to [6], put random noise images.

Other remarks:
 - I am not fully convinced by the origins of writing history. As it is quite a detail in the introduction, I would recommend supporting it with more than 1-2 papers if it is important for the authors.
 - Beware of the citations; they are surprisingly centered around one research group/company. Even if it does not break anonymity rules, I would recommend being more inclusive.
- Object-Oriented game variant. Instance classification within weakly supervised learning [7 or other paper] and/or [5] as the (first?) example in the emergent communication literature.
- As future work idea: adding extra geometric options while drawing images. (e.g. output [x, y, line/rect/circle, red/blue/green] etc.). If you do so, organize a small expo at Neurips next year :)


I made several remarks, and I am happy to increase my score further if they are addressed.


[1] Castrejon, Lluis, et al. "Learning aligned cross-modal representations from weakly aligned data." Proceedings of the IEEE conference on computer vision and pattern recognition. 2016.

[2] De Vries, Harm, et al. "Modulating early visual processing by language." arXiv preprint arXiv:1707.00683 (2017).

[3] Gatys, Leon A., Alexander S. Ecker, and Matthias Bethge. "Image style transfer using convolutional neural networks." Proceedings of the IEEE conference on computer vision and pattern recognition. 2016.

[4] Chen, Ting, et al. "A simple framework for contrastive learning of visual representations." International conference on machine learning. PMLR, 2020.

[5] Lazaridou, Angeliki, Alexander Peysakhovich, and Marco Baroni. "Multi-agent cooperation and the emergence of (natural) language." arXiv preprint arXiv:1612.07182 (2016).

[6] Bouchacourt, Diane, and Marco Baroni. "How agents see things: On visual representations in an emergent language game." arXiv preprint arXiv:1808.10696 (2018).

[7] Amores, Jaume. "Multiple instance classification: Review, taxonomy and comparative study." Artificial intelligence 201 (2013): 81-1



**Time Spent Reviewing:**

5h

---

> ### Author Response · Authors · 2021-08-10
> **Response to Reviewer pCdM**
>
> Firstly, we would like to thank the reviewer for the extensive feedback and for the great suggestions to strengthen our work. We respond to specific comments below:
>
> > While this idea is definitely not novel [1], and it has been in the air again recently [2], the quality of this work is pretty high. There is a clear effort to mix quantitative and qualitative results (with sketches). Besides, the authors did not stop their research after a few positive results, and have excellent additional experiences in section 4. Nonetheless, the paper still has some minor weaknesses that I am happy to discuss further (cf Main results).
>
> We are pleased that the reviewer appreciates the quality of this work and acknowledges the effort of providing a range of quantitative and qualitative results. We also thank them for pointing out specifically related works which we were not aware of and will happily discuss in the paper. Regarding novelty, we'd point out that perhaps a rather understated contribution of our work is that our framework can be easily extended - the idea for future experiments with coloured geometric shapes should be quite easy to add for example, and we're excited to explore that going forwards :)
>
>
> > A) The perceptual loss is one of the key components of the paper. However, I am wondering what is the impact of $L$...? Interpretability? More generally, I am missing a comprehensive ablation on the paper core elements...
>
> These are all good questions, and we've begun to gather results for a full ablation study to add to the appendices. Regarding $L$ for example, we observe that in the original game setting (STL-10, 20 lines) there is a drastic loss in communication success rate as $L$ decreases from 5: $commrate(L=5)=0.698 (\pm 0.093)$ to $commrate(L=4)=0.194 (\pm 0.099)$ (means (std.devs) from 10 seeds). Note that these results are from when $L$ is changed for both sender and receiver agents.
>
> Regarding **num_strokes (20+)**, in the original game setting played on STL-10 $commrate(numstrokes=20)=0.698 (\pm 0.093)$ (means (std.devs) from 10 seeds). We observe that increasing the number of strokes does not significantly improve the communication rate, but it reduces the noise in the communication success in some cases (for $numstrokes=30, 50$ the std is halved): $commrate(numstrokes=30)=0.71 (\pm 0.035)$, $commrate(numstrokes=40)=0.736 (\pm 0.072)$ and $commrate(numstrokes=50)=0.728 (\pm 0.031)$.
>
> Similarly, we are gathering results for an ablation study on $\lambda$, where $l=l_{game} + \lambda l_{perceptual}$. _Preliminary_ results using $\lambda \in [0,1]$ suggest, as we already showed in Tabel 1 of the paper, that adding, or in this case scaling down the perceptual loss, does not have a significant impact on the communication success. It does, however, have an impact on the *interpretability* of sketches.
>
> Finally, the idea of overlaying (or somehow otherwise showing the variation) in sketches is great - we'll work on incorporating that.
>
>
> > B) Regarding the perceptual loss, I would recommend adding a small paragraph to link it to other work that explores low/high feature levels, e.g., [1] or [2] or, more generally, to style transfer [3]. One may argue that the loss is a naive implementation of style transfer.
>
> Sure! These are great suggestions, we are happy to include that in Section 3.4.
>
> > C) What are the intuitions of using a VGG network instead of ResNet? Using Hinge-loss instead of InfoNCE loss? cf. Tab2 in [4]? Funnily, the paper is using Deep Learning techniques from 2015. Note that it does not undermine the quality of the paper.
>
> Good questions: hinge loss and the VGG16 were originally used by [A1] who used a similar game setup, but with a discrete communication channel, so it made sense to follow the same procedure. We haven't tried InfoNCE, but we did do some experiments with regular cross-entropy (e.g. assuming that the receiver predicts the logits of the softmax, and thus minimising the NNL) and didn't notice much difference. We'd also note that VGG16 is still widely used as a proxy for the early human visual system or just as a feature extractor in many recent papers [A2, A3, A4].
>
> > D) As one of the major claims of the paper is communication with humans, I would have significantly increased my score to have a human evaluation (i.e. the score of humans playing this game).
>
> Yes, good point - please see our comment in the general response to all reviewers about how we're addressing this.
>
> > E) As mentioned, the paper has many complementary experiments. Yet, I think it is also missing what I call counter-factual experiments, i.e., experiments that try to break the setting to challenge the paper hypothesis further...
>
> Again good points. We've already trained models with points instead of lines and can report on that. Communication between agents is definitely possible even when extracting as little as 10 points from an image ($commrate(npoints=10)=0.75$ in the original game setting), but the resulting *sketch* does not have any meaning to a human observer. When increasing the number of points to 50, or better 100, the communication success slightly drops to 0.71, 0.66 respectively, but object contours/shapes start to emerge in the sketches. Therefore, encoding to a cloud of points is possible but less efficient, as it requires more coordinates to be learned to create sketches that are *interpretable* (to some extent) for humans. We will happily include quantitative and qualitative results of this experiment in the appendices. Experiments with agents that have been trained on one dataset and that are then tested on noise or a different dataset are also a great idea which we can incorporate.
>
> > Other remarks:
>
> We'll revisit the origins of writing bit in the introduction. Any perceived bias in the referencing is completely unintentional and we'll certainly address that by adding further references to the broader related field. Regarding extra geometric drawing options - yes, we'd love to explore in that direction, and the nice thing is that we actually now have the technical means to do this.
>
>
> We once again thank the reviewer for their time and insights! We hope we addressed all of their remarks, and we are happy to answer any questions and to update them with additional results as we progress.
>
> [A1]: Havrylov, Serhii & Titov, Ivan. (2017). Emergence of Language with Multi-agent Games: Learning to Communicate with Sequences of Symbols. NIPS 2017
>
> [A2]: Singer, J., Seeliger, K., & Hebart, M. N. (2020). The representation of object drawings and sketches in deep convolutional neural networks. NeurIPS 2020 Workshop SVRHM
>
> [A3]: Nonaka, S., Majima, K., Aoki, S. C., & Kamitani, Y. (2020). Brain hierarchy score: Which deep neural networks are hierarchically brain-like?. Available at SSRN 3664362.
>
> [A4]: Storrs, K. R., Kietzmann, T. C., Walther, A., Mehrer, J., & Kriegeskorte, N. (2020). Diverse deep neural networks all predict human IT well, after training and fitting. bioRxiv.

---

> > ### Comment · Reviewer_pCdM · 2021-08-11
> > **Thank you for the detail rebutals**
> >
> > Thank you for your answer. I appreciate the additional results, and the comments. If you manage to run the human evaluation, I would increase my score to 9. In the meantime, 8 is already quite strong!

---

### Official Review · Reviewer_maSs · 2021-07-16

**Rating:** 8
**Confidence:** 4

**Summary:**

This paper is an exploration into getting agents to communicate through drawing. It's in the multi-agent communication literature and proposes having the agents play the traditional referential game, however the sender agent is going to draw a sketch (with differentiable rasterization) instead of communicating through a symbolic channel. The receiver then has to say which of the target images the sketch aligns with.

That agents can play this well when gradients connect them is unsurprising as we expect the result to be uninterpretable. The authors go on to show how to yield interpretable images by adding a perceptual loss based on pretrained VGG, either by ImageNet or StyleNet.

**Limitations And Societal Impact:**

Somewhat. It's a small paragraph at the end and not its own section. Also, they really should cite Pix18 as it deserves that at the least.

**Main Review:**

Quality: The paper is written well and gets to the heart of the matter.
- It sets out to show that a thing can be done and does the thing. Great.
- Fig 3 is actually what I was looking for for much of the paper and the fact that hte drawings in both the ImageNet and the StyleNet rows are barely differentiable to me suggests that the model is likely using just a couple placements / strokes to define large fetures. For example, in the last two columns I don't think the face parts of the drawings are any different but the hair parts are, and that roughly makes sense. This gives me three ideas that I would like to see explored, only one of which I think would be worth adding to this paper:
1. (Add to this if you can) Go run some human tests! See if humans can figure out what's going on in all of these experiments. Pair the agent with a human and repeat it a lot of times. Adding those scores in would be fantastic because that's what we're all after (and you too re L67-68).
2. Try to figure out what the strokes mean. I get that that's not what you're going for in this paper, but I think that that would be really insightful. I think you can do this by running causal interventions on the strokes, perhaps with a language model to learn the distribution first. CLIP would count.
3. Speaking of CLIP, try using that instead of the VGG. It's open source and been trained largely for this sort of task. It would very likely improve the results a lot.

Originality: Originality is there afaik.
- I've been in this subfield for a while and I recall people saying they were going to do this something along these lines for much of the past half decade, and then nothing (afaik) was ever done. So I'm glad to see that this direction is being explored because I think that it's fruitful to consider drawing as the communication protocol; it's much simpler and more interpretable than all of these questions that get tangled up with language.
- If though a similar paper was surfaced, then my score would likely drop. A lot of the experience of this paper is seeing someone finally do this (and it's done well).
- With respect to the last few paragraphs, I suggest looking into Pix18 (http://www.pix18.com/). It is a robot that generates art and its works have entered the public domain for sale. It also was rejected from an art competition (after first being accepted) on the basis that what it makes is not art. You should cite it as its a trailblazer in this domain.

Clarity: Overall the paper was clear.
- I do have one request though and that's to say earlier what is the actual task. It wasn't clear until page 5, Sec 3.2.2, that the receiver agent is going to get all of the real images and have to align the sketch with them to predict the maximal scoring image. This was confusing for much of the paper - just put it earlier and you'll help the reader a lot.

Significance:
- This isn't a world-bending paper, nor is it going to come across as profound to the general audience. However, it does have the potential to kick this subfield out of a rut and into a more compelling place involving drawing and pictorial approaches.

**Time Spent Reviewing:**

3

---

> ### Author Response · Authors · 2021-08-10
> **Response to Reviewer maSs**
>
> We would like to thank the reviewer for the constructive ideas that could be explored and are definitely worth adding to this paper! We respond to specific comments below:
>
> > Go run some human tests!
>
> We totally agree with this, it is definitely something that we're after! We are in the process of collecting preliminary results which we will report as soon as we have them and will happily include them in the final version of the paper. As we've described in the general response to all reviewers, we plan (at least initially) to focus on performing a pilot study in which human's play the role of the receiver (and hence have to guess which image corresponds to the artificially produced sketch). We should be able to test this under a variety of different conditions (game setups, use of perceptual loss, etc).
>
> > Try to figure out what the strokes mean.
>
> Can we ask the reviewer to expand on what they mean here? Are they asking what an individual stroke might mean in isolation? or perhaps how meaning changes as more strokes are laid down? or something different? This might not be something we can address now, but certainly, it could be added to the future work discussion.
>
> > Speaking of CLIP, try using that instead of the VGG.
>
> Great suggestion, we are trying that and we'll report results shortly.
>
> >Originality (...) I'm glad to see that this direction is being explored because I think that it's fruitful to consider drawing as the communication protocol; it's much simpler and more interpretable than all of these questions that get tangled up with language.
>
> We are encouraged that the reviewer appreciates the research direction explored here and recognises the advantages of a visual communication protocol. It is something that we have been working towards for while now.
>
> > With respect to the last few paragraphs, I suggest looking into Pix18 (http://www.pix18.com/) (...) You should cite it as its a trailblazer in this domain.
>
> This is a great suggestion, we were not aware of it. We now cite Pix18 in the last paragraphs and endeavour to also expand the limitations/impact discussion as noted.
>
> >Clarity: Overall the paper was clear. I do have one request though and that's to say earlier what is the actual task.
>
> This is a fair point. We must have overlooked that aspect being so familiar with the setup and the task usually explored in image referential games. We have now stated what the task is in the introduction. Thank you for pointing that out!
>
> >Significance: This isn't a world-bending paper, nor is it going to come across as profound to the general audience. However, it does have the potential to kick this subfield out of a rut and into a more compelling place involving drawing and pictorial approaches.
>
> We agree with the reviewer in regards to the paper's potential. We would like to see visual communication explored further and we hope that this work lays the foundations that allow for exploration of the transition from pictorial approaches to linguistic communication.
>
> We once again thank the reviewer for their time and feedback! We are happy to answer any question or provide additional explanations.

---

> > ### Comment · Reviewer_maSs · 2021-08-11
> > **Score unchanged**
> >
> > Re "Try to figure out what the strokes mean.", I'm saying to figure out what the agents are communicating to each other via the strokes. It could be a stroke in isolation (like you suggest) or it could be groups. Many language based papers in the field do this.
> >
> > I've read the other reviews and the replies, and my scores the same. There are lots of promises to improve and I think my score wouldn't change until the consensus around human tests was recognized. As it is, this is a good score and I think you have a great chance of passing the bar.

---

### Author Response · Authors · 2021-08-10
**Response on Preliminary Reviews to All Reviewers**

We want to genuinely thank the reviewers for their time and their insightful and valuable feedback. We appreciate the useful and very specific suggestions and thoughtful questions which can definitely strengthen our paper even further. We address each reviewer's comments within replies to each review. We will incorporate all feedback in the final version of the paper.

We note that all reviewers have asked for a variety of different and interesting additional experiments. It might not be possible to do all of these for this paper, however, we've tried to give an indication of what is possible (with preliminary results where we have them) in the individual reviewer responses. Regarding the much-asked-for Human evaluation, we do not want to over-promise, but we believe it should be possible to at least incorporate results from a pilot study that looks at game play when the receiver agent role is played by a human.

In our responses, we have specifically asked the reviewers about any questions we are unsure we correctly understood. It is of course possible that we have misinterpreted other comments or questions, in which case we would cordially ask the reviewers to point this out to us and we will try to reply as promptly as possible.

---

### Author Response · Authors · 2021-08-16
**Human Evaluation -- Pilot study plan**

We thank the reviewers for their updated comments. We would like to share our plan for human evaluation. We kindly ask the reviewers to add any comments if they have them; otherwise, we will proceed as follows.

We set up an initial pilot study that looks at game play success when the receiver agent role is played by a human participant. The participant will be shown a **sketch** (previously generated by a trained Sender agent during model evaluation) and will be asked to select by clicking the corresponding target image from a grid of images. Depending on the game setting, the size of the list of options will differ but it will composed of **distractors + true target image**.

- All the sketches used in this study will be generated by models trained with the same fixed random seed. Whilst there is inevitably variation in models from different seeds this will not be explored at this stage.

- As the number of game settings can quickly escalate as we vary things like #numstrokes, game loss or dataset, in this pilot study we will focus only on games played using the **STL-10 dataset**.

- Each human participant will play **30 games** from each game configuration (i.e. will be shown 30 sketches from each game configuration). The games are chosen randomly from all those possible within the STL-10 test dataset.

- *Q: Can humans "learn" to play the game"?* The principal pilot study is looking at humans' ability to play the game with an agent, with no feedback involved. The human participants will not know what the correct target was or if they guessed correctly. We also pose a slightly different question: Can humans *learn* to play the game with an agent? For this secondary study, after participants select what they believe to be the target image, they will be told if their selection was correct or not and the correct target will be indicated.

- *Q: Should there be a time limit?*  We will not impose a time limit, but we will record how long the subjects take to make their guess.

## Game Settings for Human Evaluation, no learning allowed

As mentioned already, we simply evaluate human performance on the image guessing game presented in the paper, and no feedback about their performance is given to the human participants. After selecting what they believe to be the **target** image, they will proceed to the next game (i.e. sketch).

The study will explore the following game setting:

- *original game*: human sees a sketch and has to pick the target image from a list of **10** images (9 distractor images + 1 target image).
    - We will test using 20-stroke sketches drawn by the Sender of a model trained using $l=l_{game}$ only;
    - We will test using 20-stroke and 50-stroke sketches drawn by the Sender of a model trained with $l=l_{game} + \lambda l_{perceptual}$, where $\lambda=1$;
- *object-oriented different*: human sees a sketch and has to pick the target image from a list of 10 images, each corresponding to a different class in STL-10. In this case, the list will not contain the exact image that was used to draw the sketch, but will contain a different image from the same class; that is considered the target.
    - We will test using 20-stroke and 50-stroke sketches drawn by the Sender of a model trained with $l=l_{game} + \lambda l_{perceptual}$, where $\lambda=1$;


In total there will be 5 possible settings, 30 games for each, hence 150 games to be played by each human participant. We will report means and standard deviations for human performance for each of the settings.

## Game Settings for Human Evaluation, learning allowed from feedback

We aim to also run exactly the same set of experiments but with feedback given immediately after a guess is made. The human participants will be disjoint from those asked to participate in the first study.

---

### Author Response · Authors · 2021-08-23
**Results of Pilot Study (Human Evaluation)**

We would like to present the results of the human evaluation pilot study. We followed the plan described in our previous comment and we now present results from the two disjoint groups (with/without feedback). For this pilot study, we collected results from 6 participants per group.

To reiterate the experiment, each participant was provided with a set of 5 unique URLs (each corresponding to a game setting, as per our previous comment) and they were asked to play the role of the Receiver. For each game setting, the human participant had to play 30 games, in other words, match 30 sketches to the target image. Each human participant played a total of 150 games, and the total amount of data collected in the pilot study corresponds to 1800 games.

We first highlight and discuss our main observations on the results of this pilot study. The full results for the two disjoint study groups are presented below.

> Does the addition of perceptual loss (i.e. $l=l_{game} + l_{perceptual}$) give statistically significant improvement over games which use only the hinge loss ($l=l_{game}$)?

All participants were asked to play the original game (numstrokes=20) with $l=l_{game}$ and also with $l=l_{game} + l_{perceptual}$. In the group without feedback,  the averaged communication success for the two scenarios were $commrate=0.083\ (\pm0.054)$, and $commrate=0.383\ (\pm0.025)$ respectively. Similar results were observed in the group which was provided with feedback: for the game without perceptual loss $commrate=0.139\ (\pm0.030)$; with perceptual loss, the communication success improved to $commrate=0.378\ (\pm0.074)$. Performing t-tests between the averaged communication rates within each group, with and without perceptual loss, resulted in P values less than 0.0001 which indicates that the perceptual loss leads to a statistically significant improvement in humans’ ability to play the game with the agent.

> Can communication with an agent be improved if the human participants are allowed to learn via feedback?

T-tests run between the averaged communication success rates of the same game setting in the group with feedback versus the one without feedback, do not show a statistically significant improvement when participants are allowed to learn from feedback, except for the original game with $l=l_{game}$ only. As we expected, participants in both study groups had the lowest scores in this game across all tested settings: without feedback the averaged $commrate=0.083\ (\pm0.054)$; with feedback, $commrate=0.139\ (\pm0.030)$. The sketches drawn by a Sender agent pretrained without the perceptual loss are not "constrained" to resemble the target image, hence they are the least interpretable. However, the two-tailed P value between the two groups performance in this setting was less than 0.0001 which suggests that feedback can lead to a statistically significant improvement when the sketches are not visually interpretable. Still, this is by far the worst communication scenario, this is also indicated by the amount of time the participants spent on average on this game which is higher than in other settings, the majority taking between 1 minute and 2 minutes and 30 seconds per sketch.

In the future, it would perhaps be interesting to explore if humans could learn with feedback if they were to play more games with feedback; the 30 games per setting used in this experiment is possibly too little to allow a human player to learn the strategy used by the agent.

> Does the number of strokes influence human performance?

We tested the original game and the object-oriented game setup, each with 20 and 50 strokes. The results indicate that in both settings, a higher number of strokes leads to better communication. However, in the group without feedback, the mean communication rate was similar for the original setting with 20 strokes and with 50 strokes. The same game setting tested by people with feedback however, showed a small increase in overall communication success. One should take into account that in this game setting, the human player might have to choose between more images from the same class. For an artificial agent, this can be an easy task. However, we might envisage a scenario in which other characteristics of a drawing would be included, such as colour, which might help the human differentiate between multiple instances from the same class. For example, think of 3 different species of birds, which could all be represented by some very general sketch, but could become distinctive if the colour were to be included.

In the object-oriented game setting, the gap between 20-stroke and 50-stroke games is a bit more significant. The average success rates for the group without feedback are: $commrate(numstrokes=20)=0.239\ (\pm0.062)$ compared to $commrate(numstrokes=50)=0.389\ (\pm0.099)$. Similar improvement can be seen in the group provided with feedback: from $commrate(numstrokes=20)=0.322\ (\pm 0.105)$ to $commrate(numstrokes=50)=0.417\ (\pm 0.129)$.


>Are humans better at determining the broader class of a sketch than at recognising the specific instance?

In the original game setting, it is possible to encounter distractor images from the same class as the target. In addition to the **$commrate$** measure, which shows the overall success of an agent (human in this case) selecting the correct target image, we also compute the **$classcommrate$** which calculates the overall success of an agent selecting an image from the same class as the true target. T-tests run between human $commrate$ and human $classcommrate$ in the original game settings showed a statistically significant difference in both study groups. Humans are significantly better at understanding the broad class than they are at determining a specific instance based on the sketch in the games where there are multiple targets of the same class. This effect is possibly weakened by an increase in the number of strokes, however, as evidenced by a consistent lowering of statistical significance.



## Full Results: Human Evaluation, no learning allowed

The participants in the first study group played the games without any feedback on their performance, hence there was no opportunity to learn.

For each game setting, we report communication success - **$commrate$** (i.e. human choosing the correct target image) mean and standard deviation over participants. For comparison, we also show the model's performance based on its guesses for the same games played by the humans. In addition, for the games played in the original setup in which it is possible to encounter distractor images from the same class as the target, we also report **$classcommrate$** which takes into account if the human chose an image from the same class as the target, but not necessarily picking the correct image within that class.



| Game | Loss | Lines | Agent commrate | Human commrate | Human classcommrate |
| ----------- | ----------- | ----------- | ----------- | ----------- | ----------- |
| original | $l=l_{game}$ | 20 | $1.00$ | $0.083\ (\pm0.054)$ | $0.150\ (\pm0.025)$ |
| original | $l=l_{game} + l_{perceptual}$ | 20 | $0.933$ | $0.383\ (\pm0.025)$ | $0.556\ (\pm0.071)$ |
| original | $l=l_{game} + l_{perceptual}$ | 50 | $1.00$ | $0.372\ (\pm0.059)$ | $0.478\ (\pm0.074)$ |
| oo diff | $l=l_{game} + l_{perceptual}$ | 20 | $0.833$ | $0.239\ (\pm0.062)$ | $0.239\ (\pm0.062)$ |
| oo diff | $l=l_{game} + l_{perceptual}$ | 50 | $0.900$ | $0.389\ (\pm0.099)$ | $0.389\ (\pm0.099)$ |


## Full Results: Human Evaluation, learning allowed from feedback

The participants from the second group were tested on the same set of experiments as the first group, and the same metrics are reported. The people in this group, however, were given feedback immediately after they made a choice, hence there was an opportunity to learn the communication protocol of the Sender agent.


| Game | Loss | Lines | Agent commrate | Human commrate | Human classcommrate |
| ----------- | ----------- | ----------- | ----------- | ----------- | ----------- |
| original | $l=l_{game}$ | 20 | $1.00$ | $0.139\ (\pm0.030)$ | $0.200\ (\pm0.051)$ |
| original | $l=l_{game} + l_{perceptual}$ | 20 | $0.933$ | $0.378\ (\pm0.074)$ | $0.544\ (\pm0.079)$ |
| original | $l=l_{game} + l_{perceptual}$ | 50 | $1.00$ | $0.428\ (\pm0.076)$ | $0.522\ (\pm0.037)$ |
| oo diff | $l=l_{game} + l_{perceptual}$ | 20 | $0.833$ | $0.322\ (\pm0.105)$ | $0.322\ (\pm0.105)$ |
| oo diff | $l=l_{game} + l_{perceptual}$ | 50 | $0.900$ | $0.417\ (\pm0.129)$ | $0.417\ (\pm0.129)$ |


Finally, we want to thank the reviewers again for their valuable feedback. We'll continue to build upon the ideas and results discussed here and to further analyse the results as we write them into the paper.

---

> ### Comment · Reviewer_9t7y · 2021-08-25
> **Response**
>
> These are cool results! To clarify, `commrate` is human accuracy at selecting the right target given an agents' drawing, where chance performance is 10%?

---

> > ### Author Response · Authors · 2021-08-25
> > **commrate**
> >
> > Thank you! Yes, exactly. (Human) commrate is the human accuracy, and because of the way we set up the experiments in this pilot, the chance accuracy is 0.1 (e.g. 10%) in all game settings.

---

> > > ### Comment · Reviewer_9t7y · 2021-08-25
> > > **Thanks**
> > >
> > > Thanks! I’ve updated my score.

---

> ### Comment · Reviewer_maSs · 2021-08-25
> **Nice!**
>
> Thanks for adding these in. Updating score.

---

### Decision · Program_Chairs · 2021-09-27

**Decision:**

Accept (Oral)

**Comment:**

This is a fun paper! The basic ideas of communication by sketching, emergent communication, and indeed emergent communication via sketches have been floating around recently, but the focus on interpretability of the emergent sketches is a valuable addition. As the reviewers indicate the addition of human evaluations is very nice, and some space should be given in the revision to interpreting these new results.

One comment from me: I would love to see a little discussion of the connections to human communication with sketching, eg "Pragmatic Inference and Visual Abstraction Enable Contextual Flexibility During Visual Communication." J. E. Fan, R. D. Hawkins, M. Wu, & N. D. Goodman. (2019). Computational Brain & Behavior.